# Scientific modelling can be accessible, interoperable and user friendly: A case study for pasture and livestock modelling in Spain

Alba Marquez Torres[1]*, Stefano Balbi[1,2], Ferdinando Villa[1,2]

1 Basque Centre for Climate Change, Bilbao, Biscay, Spain, 2 Ikerbasque Foundation for Science, Bilbao, Biscay, Spain

* alba.marquez@bc3research.org

## Abstract

This article describes the adaptation of a non-spatial model of pastureland dynamics, including vegetation life cycle, livestock management and nitrogen cycle, for use in a spatially explicit and modular modelling platform (k.LAB) dedicated to make data and models more interoperable. The aim is to showcase to the social-ecological modelling community the delivery of an existing, monolithic model, into a more modular, transparent and accessible approach to potential end users, regional managers, farmers and other stakeholders. This also allows better usability and adaptability of the model beyond its originally intended geographical scope (the Cantabrian Region in the North of Spain). The original code base (written in R in 1,491 lines of code divided into 13 files) combines several algorithms drawn from the literature in an opaque fashion due to lack of modularity, non-semantic variable naming and implicit assumptions. The spatiotemporal rewrite is structured around a set of 10 namespaces called PaL (Pasture and Livestock), which includes 198 interoperable and independent models. The end user chooses the spatial and temporal context of the analysis through an intuitive web-based user interface called k.Explorer. Each model can be called individually or in conjunction with the others, by querying any PaL-related concepts in a search bar. A scientific dataflow and a provenance diagram are produced in conjunction with the model results for full transparency. We argue that this work demonstrates key steps needed to create more Findable, Accessible, Interoperable and Reusable (FAIR) models beyond the selected example. This is particularly essential in environments as complex as agricultural systems, where multidisciplinary knowledge needs to be integrated across diverse spatial and temporal scales in order to understand complex and changing problems.

## Introduction

Extensive farming, when paired with the conservation of natural vegetation, has historically been capable of sustaining food production in agricultural areas while maintaining ecosystems in good condition [1–3]. Since the 1950s, the increase of labour costs and beginning of widespread mechanisation and fertilizer application in the developed countries [4, 5] led to important changes such as the intensification of land use and the expansion of farming scale. This

**Data Availability Statement:** All relevant data and code are within the paper and its Supporting Information files. The dataset used in PaL model is in the code, inside the resources folder. All the data have been generated by us and are available on

Zenodo. (DOI: 10.5281/zenodo.7533051). PaL model and its dataset is under GNU Affero General Public License v3.0 license. Base model on which we have based to adapt it to the PaL semantic model in k.LAB are on Zenodo (DOI: 10.5281/zenodo.6786418). The model is developed by PhD. Juan Busqué and it is under GNU Affero General Public License v3.0 license.

**Funding:** AMT: - Spanish Ministry of Economy and Competitiveness MINECO (BC3 María de Maeztu excellence accreditation MDM-2017-0714). https://www.aei.gob.es/convocatorias/buscador-convocatorias/apoyo-centros-excelencia-severo-ochoa-unidades-excelencia-2 AMT, SB, FV: - Interreg Atlantic Area Programme 2014-2020 (EAPA 261 - 2016 ALICE). https://www.atlanticarea.eu/ https://project-alice.com/es/alice-project/ - Basque Government (BERC 2018-2021 program) https://www.euskadi.eus/ayuda_subvencion/2017/berc/web01-tramite/es/ SB, FV: - Ikertzaile Doktoreentzako Hobekuntzarako doktoretza-ondoko Programme. https://www.euskadi.eus/ayuda_subvencion/2021/posdoc-berriak-orokorra/web01-tramite/es/ The funders had no role in study design, data collection and analysis, decision to publish, or preparation of the manuscript.

**Competing interests:** The authors have declared that no competing interests exist.

paradigm shift benefited from the European Union's Common Agricultural Policy (CAP) subsidies scheme. These policies simultaneously contributed to disincentivizing low-input land uses, causing land abandonment and afforestation in extensive agricultural areas, while also decreased agricultural commodity prices due to overproduction of intensive farming [6, 7].

Today, multiple human activities, such as urban development and tourism, are adding further pressures to ecosystems in addition to the increased productivity of intensive agriculture. These activities are moving pasture from mountain areas [8] to more accessible locations closer to urban centres [9]. Agricultural intensification in concentrated areas is threatening ecological sustainability and the provision of ecosystem services [10, 11]. Such pressures are leading to ecosystem degradation by reducing biodiversity and threatening species linked to low-intensive agricultural production [12–14], and by depleting plant resources, increasing contamination by leachate and soil erosion [15–19].

At the same time, farmland abandonment in rural areas can cause: (i) loss of woodland clearings, (ii) increased fuel loads and fire hazards and (iii) negative impacts on biological diversity [20, 21]. The improvement of farmers' socio-economic conditions, extensive farming evolution and the balance with the environment require more efficient use of pastoral vegetation, including proper livestock management (grazing rotations by species and across time and space [22]) and the controlled use of fire to preserve pasture availability [23, 24]. The lack of quantitative tools for the analysis of such processes has been a major limitation for smarter and more sustainable management of mountain pastureland [25].

Agricultural production systems have benefited from technological advances primarily developed for other industries such as mechanization, synthetic fertilizers, genetic engineering and automation. The information age brings new technology that can transform agriculture to low-input, high-efficiency and sustainable systems [11, 26], such as cloud computing, remote sensing and artificial intelligence [27–29]. The agricultural industry is now capable of gathering more comprehensive data on production variability across both space and time [30]. Data and models can play an important role in sustainable agriculture, optimizing resources, providing key spatial-temporal information and identifying the most appropriate and effective practices for better management [31].

One of the main issues preventing the full use of these new technologies in agricultural modelling arises from the multidimensional nature of needed data and models that are produced by different scientific domains from climatology to ecology and social sciences [32]. Although an agricultural system can be designed for a specific purpose, such as crop production or animal breeding, understanding it requires knowledge from diverse fields (e.g., agricultural production, natural resources and human factors) [33, 34]. These components cannot be studied in isolation [35], since they interact with each other and with their environment [36].

The Puerto model [37, 38] was created in response to some of the above-mentioned agricultural systems challenges, combining well–established knowledge and algorithms from international scientific research [39–46]. The Puerto model was developed at the Centre for Agricultural Research and Training of Cantabria (CIFA) as part of its research on the structure, growth and utilization of pastures in the Cantabrian rangeland. Puerto is an empirical dynamic model based on established biophysical relationships and constants between vegetation's life cycle (including growth, senescence, vegetation death and litterfall), livestock grazing process (livestock ingestion, digestion, excretion and weight change) and the nitrogen cycle (nitrogen uptake, soil cycling and leaching). It evaluates existing nitrogen and grazing imbalances (under- or overgrazing) and their relationship with animal productivity. Puerto's four main goals are to: (i) provide a tool to support pastoral management; (ii) quantify and assess grazing system and nitrogen cycle imbalances; (iii) enable managers to develop strategies to resolve imbalances; and (iv) visualize the effects of management actions through scenarios.

This model has proven to be a valuable tool for modelling pastureland in Cantabria and was used in several regional projects [47–49] at different temporal and spatial scales. Although its reliability and usefulness have been validated and improved over the years, this model is essentially inaccessible to a non-initiated programming audience, and Cantabrian land managers must rely on technical consultancies to use it. Further, Puerto has always been used in isolation, never contributing to more comprehensive computational modelling dataflows. We argue that these limitations arise from three choices made in Puerto's modelling philosophy, which are typical to modern environmental modelling:

1. the model's interface is not user friendly, it is coded in R and is only usable by advanced R users, with each run requiring the modification of source files to point to input data;

2. it is monolithic and cumbersome (1,491 lines of code divided into 13 script files and linked to 19 input tables), which makes understanding of its computational dataflows difficult;

3. it lacks transparency in the definition of multiple parameters, which lack semantics and appear as acronyms defined as fixed values in the code.

These limitations are common practice in most current scientific modelling exercises, which are not developed as Findable, Accessible, Interoperable and Reusable (FAIR) scientific artifacts [50–53]. At the same time, the importance of accessibility, interoperability and reusability of models and resources is increasingly recognized by modelling communities. While novel approaches are available to facilitate that [54–60], none has yet reached the necessary levels of practicality, generality and community acceptance to make a dent into a still widespread model and data curation malpractice.

The ALICE project (https://project-alice.com), which started in 2017 and is ongoing, explored the problem of data and model integration in four case studies within the Atlantic European Region (located in Spain, Portugal, France and North Ireland). The aim of this article is to demonstrate the implementation of Puerto in one of the ALICE project case study into a semantic-first modelling approach, which aims to better achieve the FAIR criteria. This redesign makes the models, from now on referred to as the Pasture and Livestock (PaL) namespace (s), and their results more accessible to end users such as farmers and policy-makers.

PaL [61] is written in k.IM, a semantic modelling language designed for the k.LAB modelling platform. K.LAB uses artificial intelligence, and in particular semantics and machine reasoning, for the integration of data and models [62, 63]. PaL is part of ARIES (ARtificial Intelligence for Environment and Sustainability), the best known application of k.LAB [64, 65]. ARIES is used by an international and multidisciplinary community, through different a web applications linking, synthesizing and providing easy access to integrated knowledge to address a wide range of sustainability problems [66]. In this article, we describe the PaL implementation and its application to a study area in Northern Spain.

In the methods section, we describe the key requirements and distinctions of the semantic modelling approach as applied to Puerto and PaL. Our results compare the outputs of Puerto and PaL when applied to a region in eastern Cantabria and illustrate key end-user features of the k.LAB modelling environment. Finally, our discussion and conclusions describe implications of this approach for environmental modelling more generally and agricultural modelling specifically.

## Materials and methods

### Study area

The study area was selected to match the location where the original model has been most frequently applied. The Pas, Miera, and Ason watersheds ($43°20'36''N$, $3°44'28''W$) are adjacent

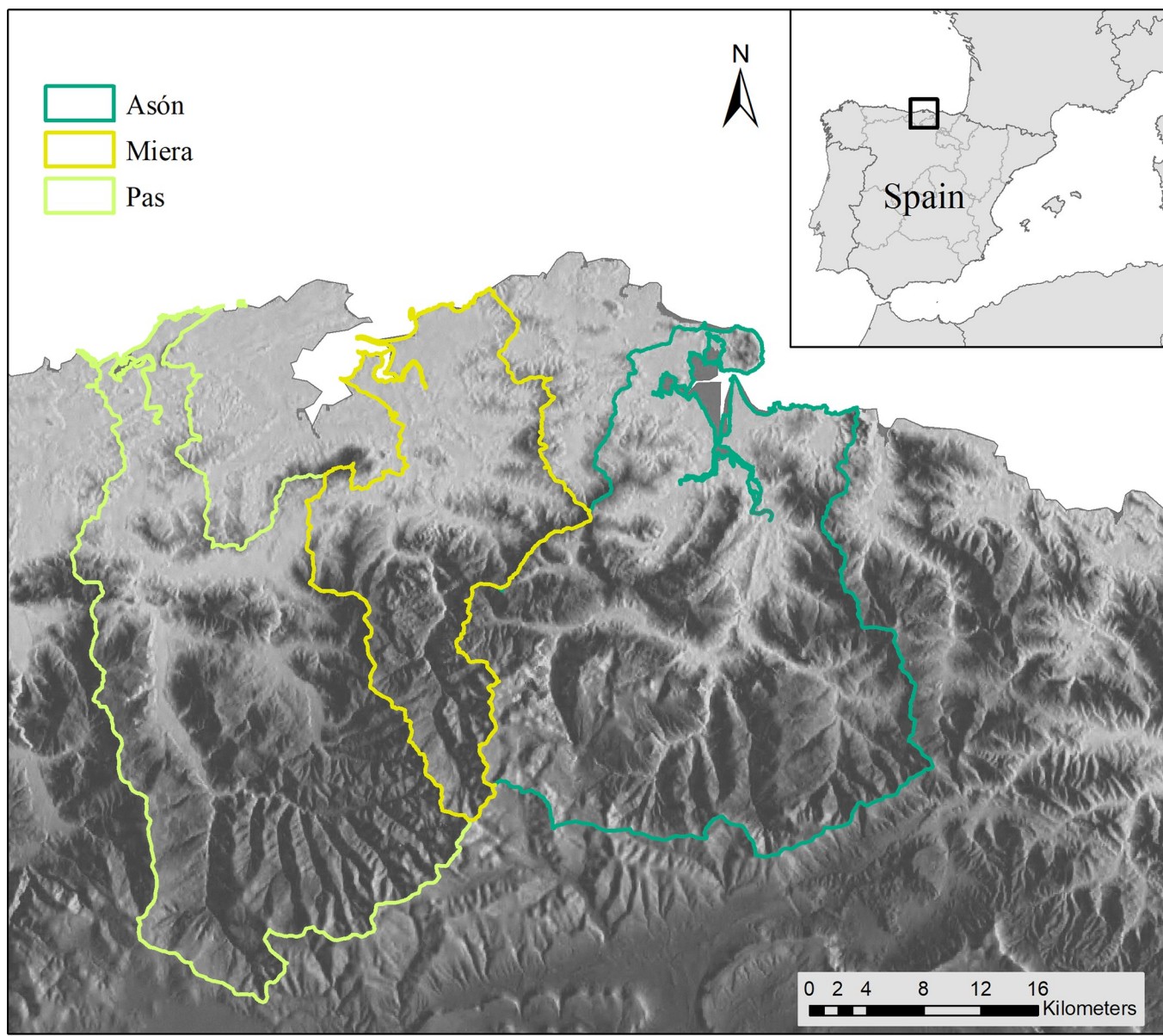

**Fig 1. Location of the case study, the Pas, Miera and Ason in watersheds in northern Spain.** Reprinted from [67] under a CC BY license, with permission from ISPRS Journal of Photogrammetry and Remote Sensing, original copyright 2014.

to the Cantabrian mountain range in the eastern Cantabria region, covering a terrestrial, riverine and estuarine system of 173,700 ha (Fig 1). This study area, with its river basins draining into the Cantabrian Sea, has a temperate hyper-oceanic climate, defined mainly by mild temperatures and high humidity due to regular precipitation and fog. Although the average annual temperature is 14˚C, snow is common in the mountains from late autumn to early spring.

This unique landscape is a product of the combined use of fire and livestock grazing for over 400 years [68]. As a consequence, almost 75% of the landscape consists of managed grasslands and shrublands, relegating mature forests to headwater basins and marginal lands with low agricultural value on steeper slopes. The pastoral lands are dominated by nine pastureland types and multiple livestock types, including cattle and mares (Fig 2).

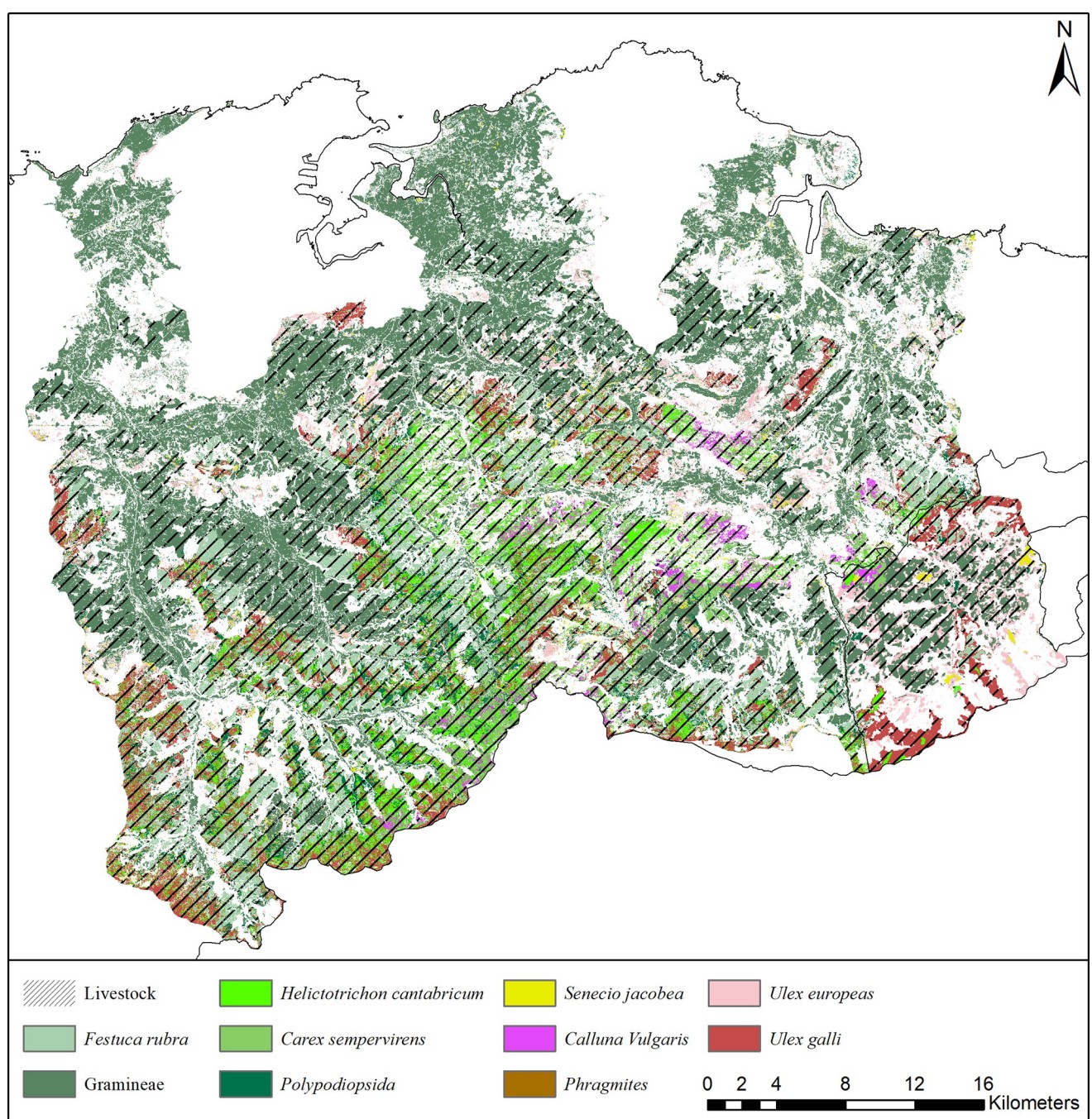

**Fig 2. Distribution of livestock and pastureland types in the case study.** Reprinted from [69] under a CC BY license, with permission from GDAM (Global Administrative Areas), original copyright 2018–2021.

There are three climatic sub-regions influenced by the mountain ranges (including the "Picos de Europa" mountain range) and the ocean. The coastal zone, which is under high human pressure, has widespread grasslands and eucalyptus plantations (*Eucalyptus globulus*) on gentle slopes. The central part is the most rugged, with elevation ranging between 100–1200 m a.s.l., dominated by semi-extensive pastures grazed by livestock. Large areas are occupied by *Ulex europaeus*, *Erica tetralix*, *Pteridium aquilinum* or *Carex asturica* and productive

plantations of *Pinus radiata*. The mountain ranges in the south, with steep slopes and a more complex management, have a great diversity of plant communities used by livestock.

## Model description

**The Puerto model.** Puerto's main code can be divided into four components. The first component consists of climate, topography and soil, which affect vegetation growth and livestock ingestion of forage. The second part captures the entire life cycle of vegetation including growth, senescence and litterfall. The third component focuses on the nitrogen cycle, including mineralization, plant reabsorption and leaching. Finally, the last component describes grazing, ingestion, weight variation, and excretion of manure and urine of livestock.

Puerto needs a substantial set of input data and parameters to be initialized, derived from literature or field measurements, related to vegetation, soil, climate, the nitrogen cycle and, optionally, livestock management. In total, it requires 56 data inputs as tables and 27 constant parameters. After initialization, it executes dynamic transitions over a modeller-defined temporal horizon, by daily timestep for entire years, simulating the management of pastoral systems. Outputs are produced in tabular format for each management unit of pastureland, and can be associated to their polygons in any Geographical Information System (GIS).

**Pasture and livestock (PaL).** The Pasture and Livestock (PaL) namespaces provide an integrated modelling framework for the Puerto model designed to better adhere to the FAIR Principles while making the model more accessible for nontechnical users. They operate under the k.LAB open-source software platform and k.IM semantic annotation and modelling language (Table 1) [62, 63] the only programming language using semantics as the primary organizational principle. The k.LAB platform connects a network of data, models, and semantic resources distributed globally on the semantic web. The code of PaL in k.IM language and the instructions to run the models are available in its online open repository [61].

Semantics are used to annotate all resources (i.e., data and model components) in PaL namespaces, using a well-established and expert-vetted vocabulary [70]. The concepts used to build the model components and to represent data are not built specifically for a model, but come from a shared, network-accessible worldview which provides uniform definitions encompassing concepts and the relationships between them. The use of semantics to describe data and models enables an artificial intelligent algorithm to build meaningful connections between inputs and outputs by making inferences and ranking each model component for the best fit to the concepts required as input. Any resource available in k.LAB can be automatically and accurately interpreted by a receiving system [71] as a response to a query. Such a modelling approach is modular by design, parsimonious and logically consistent, which makes the knowledge contained in the resources unambiguously and more transparently sharable while making the model more accessible for non-technical users. By providing a web-based query tool with intuitive spatial and temporal context selection (k.Explorer), the scientific information in models and data can be displayed in understandable and accessible fashion, without compromising on rigor and machine-readability of results.

PaL is structured into 10 k.IM code files (namespaces), which integrate multiple data and models related to climatic growth limitations, vegetation's life cycle, livestock grazing and the nitrogen cycle. PaL generates spatially explicit outputs at user-specified temporal and spatial scales. In case the user does not want to change the output characteristics, a set of default output properties is defined. These features are: spatial resolution of 50 meters, daily time step, time period between 2018 and 2050. Each model finds its input data on the network, previously annotated from international and recognized data providers from regional to global scale and from the literature; the choice of data is done by the k.LAB AI based on fit to the

**Table 1. Comparison between R and k.IM language for the "Potential above ground biomass caused by growth" model.**

| Model of Potential above ground biomass caused by growth | |
|---|---|
| **Language** | **Code** |
| R | setkey(Fhijt,com2);setkey(pl1$B3,com) T1<-pl1$B3[Fhijt][,.(IDMancha,com = i.com,com2 = com,t, diay,FT,FR,FH,FTRH,xi,ph,prPe,crecpot = FTRH*xi*ph)] |
| k.IM | **model** im:Potential ecology:AboveGroundBiomass **caused by** biology:Growth **in** g/m^2 'AboveGroundBiomass caused by Potential Growth' **observing** im:Maximum ecology:Biomass **caused by** biology:Growth **in** g/m^2 **named** xf, **percentage of** ecology:Vegetation biology:Growth **caused by** *ecology:VegetationLimitingFactor* **named** ftrh, **occurrence of** ecology.incubation:PhenologyActivity **named** ph **set to** [xf*ftrh*ph]; |

context and the scale chosen by the user. The user can also provide data to override any of the PaL components, be them input datasets or computational logics for each of the concepts involved in the model. Outputs include multiple open-source models, algorithms and spatial outputs of primary interest to pastureland managers. For example, selected results include the amount of above and below ground biomass of vegetation, concentration of nitrogen leaching or livestock weight gain. These outputs can be used for quantitative analysis of pastureland sustainability (or assessment of farmland requirements and tradeoffs). The set of PaL models are divided into 10 thematic namespaces that describe the interactions between vegetation, animals and their environment (Fig 3, Table 2).

Each namespace, in turn, is composed of several model components that each describe one concept involved in the PaL logical structure, for a total of 198 models that are logically consistent, self-contained and can run independently. The dependencies between models are defined at the purely logical level as concepts, and are resolved at the moment of execution by the k. LAB engine: if needed, the modeller can influence the choice using well-defined scoping rules. When dependencies cannot be satisfied within the same namespace or project, or within user-provided data and models, the k.LAB engine will look for ways to satisfy them by looking up models from the network and ranking them for appropriateness to the context. The ability to access the entire k.LAB semantic web enacts a fully distributed, interoperable chain of computation that minimizes the effort involved in producing results without compromising on quality, transparency or traceability.

In this particular implementation, all models are deterministic, using equations and look-up tables derived from the literature and expert knowledge. For example, the simplest namespace, the Radiation namespace (Table 2), is composed of the "Solar Radiation over Vegetation" model, which includes three different component models. Each of these sub-models generates an output and, at the same time, is interoperable with others to generate more complex models, such as the "Solar Radiation limiting factor causing Vegetation Growth" model (Fig 4).

In addition, each of these models interact with other namespaces. For example, Fig 5 shows the model of "*Nitrogen in living aboveground biomass caused by cattle solid manure*" from the 'Excretion' namespace (Table 2), which is composed of three different models that are developed within other namespaces. For example, "*Proportion of Living AboveGround Biomass in Cattle Digestion*" is located within the "*Livestock mass*" namespace while '*Proportion of Nitrogen in Living AboveGround Biomass* 'is in the "*Nitrogen*" namespace and '*Living AboveGround Biomass causing Cattle Ingestion*' in "*Ingestion.*"

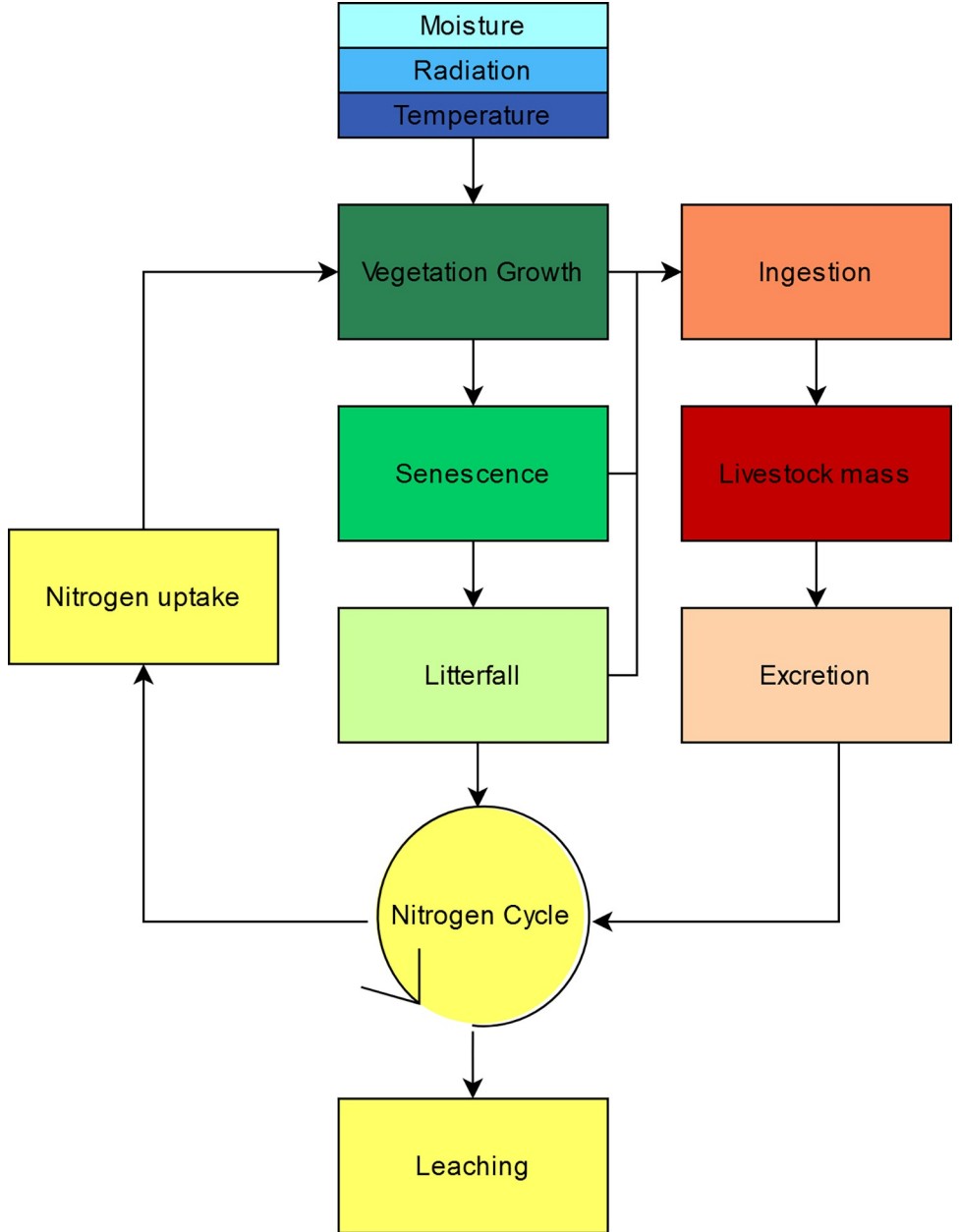

**Fig 3. Dataflow of PaL namespaces related to climatic growth limitations, vegetation life's cycle, livestock grazing and nitrogen cycle.**

In this way, each namespace is composed of models that can run independently, unlike Puerto's original monolithic structure. This semantic-driven interoperability allows each model to interoperate with models from the same namespace or from different ones, according to the projects available in the k.LAB resource network and ARIES project. For example, the nitrogen leaching model can interoperate with a runoff model from an independently developed hydrological modelling project, automatically connecting knowledge across these projects. Consistency is maintained through the semantic infrastructure, generating an integrated response to user queries and scenarios.

**Table 2. Description of PaL namespaces related to climatic growth limitations, vegetation life cycle, livestock grazing and nitrogen cycle.**

| Namespace | Description |
|---|---|
| Moisture | All processes involving the limitation of vegetation growth due to soil moisture. |
| Radiation | All processes involving the limitation of vegetation growth due to solar radiation. |
| Temperature | All processes involving the limitation of vegetation growth due to atmospheric temperature. |
| Vegetation Growth | Calculation of potential and actual vegetation growth depending on limiting abiotic factors |
| Senescence | Senescence process and quantity of the remaining, living biomass |
| Litterfall | Process related to dead plant material (harvesting, litterfall and dead biomass) |
| Ingestion | All processes related to grazing and digestion. |
| Livestock mass | Set of models related to livestock weight change. |
| Excretion | The process of livestock solid and liquid manure. |
| Nitrogen | Nitrogen concentration and nitrogen proportions in the N-cycle (including leaching and nitrogen uptake) |

PaL models use spatially explicit data (raster and vector) and look-up tables as input files. Most of the data come from field-validated expert knowledge [38, 48, 49] and the raster dataset created for the ALICE case studies [72], including for instance the raster dataset of main pastureland species or the daily weather reconditions. Moreover, open-source data from global to local scale with different temporalities can complement the model when local parameters are missing, such as the raster data describing soil texture [73]. Based on the user-defined spatial and temporal context, k.LAB changes the spatial resolution and harmonizes the spatial reference and the units of input data on the fly. Each input dataset can thus have different spatial and temporal resolution, which are automatically mediated by the system based on a given user query.

To use the model in dynamic mode, PaL requires climate data for the entire model timeline. The rest of the inputs are only needed at initialization, because PaL generates the transitions based on the declared algorithms.

## Results

The main result of PaL, the k.LAB-compatible recoded version of Puerto, is the ability to calculate any of the 198 component models independently and quickly; making them reliably available to stakeholders with minimal work (depending on the model, from seconds to 6 minutes at 50 meters' spatial resolution). The results generate parameters with self-explanatory variable names, thanks to the k.IM semantic language (Table 1). Both the data sources and the algorithms used as inputs for the results are automatically generated, and are publicly available and downloadable, giving the users additional information to interpret and communicate model results and maintain quality control (see "End-user features" below).

In the following sections, we describe the main outputs of each PaL namespace for the Cantabrian Pas, Miera, and Ason watersheds, thus emphasizing the importance of taking a systems approach in agricultural modelling. The main outputs are temporally explicit raster data produced on demand for the context of analysis (including the selected spatial and temporal scales). As the graphical outputs of the Puerto model are limited, predetermined and based on a monolithic code structure, it is difficult to directly compare all the PaL model results with those of the original Puerto model. However, we can validate some of the PaL results that directly match the final Puerto outputs. For this, the Puerto results had to be post-processed and spatialized using the management units' polygons (S2 Appendix in S1 File).

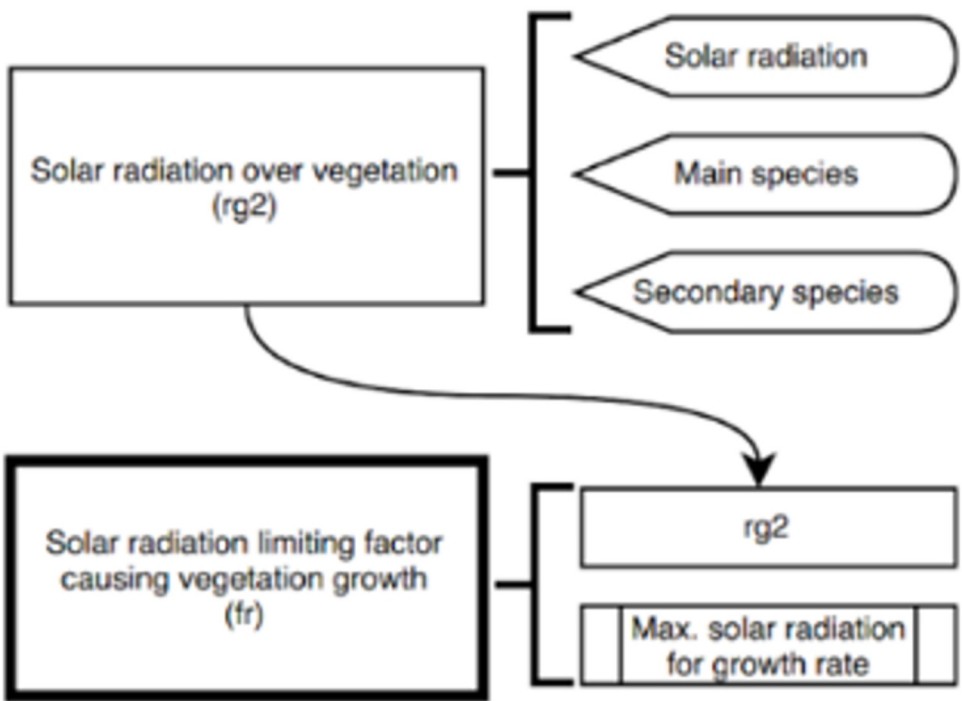

**Fig 4. Dataflows of "Solar Radiation over Vegetation (rg2)" and "Solar Radiation limiting factor causing Vegetation Growth (fr)".** The figure shows the interaction between them and the models that act as inputs. A total of six models are involved in this process.

## Model outputs

The most relevant models for stakeholders in each PaL namespaces are described below as an example, such as: limiting factors of vegetation growth, potential and actual vegetation growth, livestock weight variation and leached nitrogen. The entire list of the models is available in the Supporting Information, S1 Appendix in S1 File. The following PaL models outputs have been run at the default spatial resolution of 50 meters using mean climate values for May 2018. Although k.LAB can run these models in any temporal context, in the following example we focused on May 2018 because it was the starting date of the vegetation distribution map,

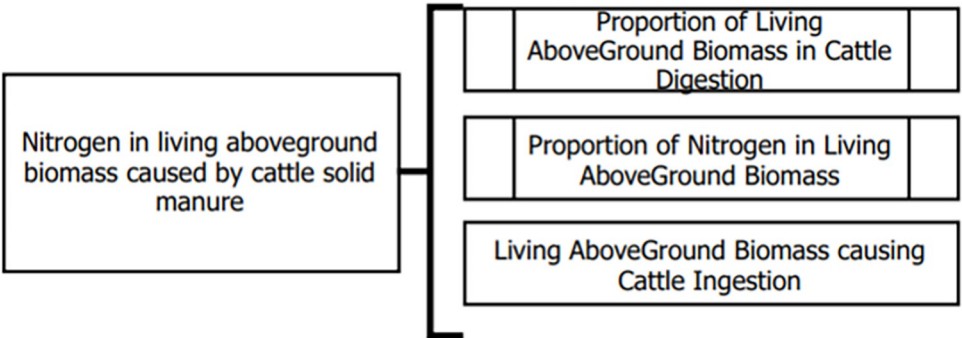

**Fig 5. Dataflow of "Nitrogen in living above ground biomass caused by cattle solid manure" model (excretion namespace).** "Nitrogen in living above ground biomass caused by cattle solid manure" is composed of three models related to namespaces of livestock mass, nitrogen and ingestion.

according to our data availability. Other data resources used in PaL models are related to weather, topography or livestock distribution and did not have the same limitation.

## Factors limiting vegetation growth

The factors limiting vegetation growth (S1 Appendix: S3C Table in S1 File) are captured in four main models (Fig 6): Moisture (S1 Appendix: S2 Fig and S2A-S2C Table in S1 File), Radiation (S1 Appendix: S3 Fig and S3A-S3C Table in S1 File), Temperature (S1 Appendix: S4 Fig and S4A-S4C Table in S1 File) and Nitrogen (S1 Appendix: S11A, S11B and S11A-S11C Table in S1 File). These dynamic models quantify climatic and soil conditions' control of potential vegetation growth. Vegetation growth follows an annual cycle influenced by seasonal patterns and extreme weather events. These models thus depend on time and can help to forecast changes in vegetation behaviour with climate change, as seasons shift and extreme events become more frequent. Moreover, factors limiting vegetation growth are affected by the spatial distribution of vegetation, which is influenced for example by the presence of mountain

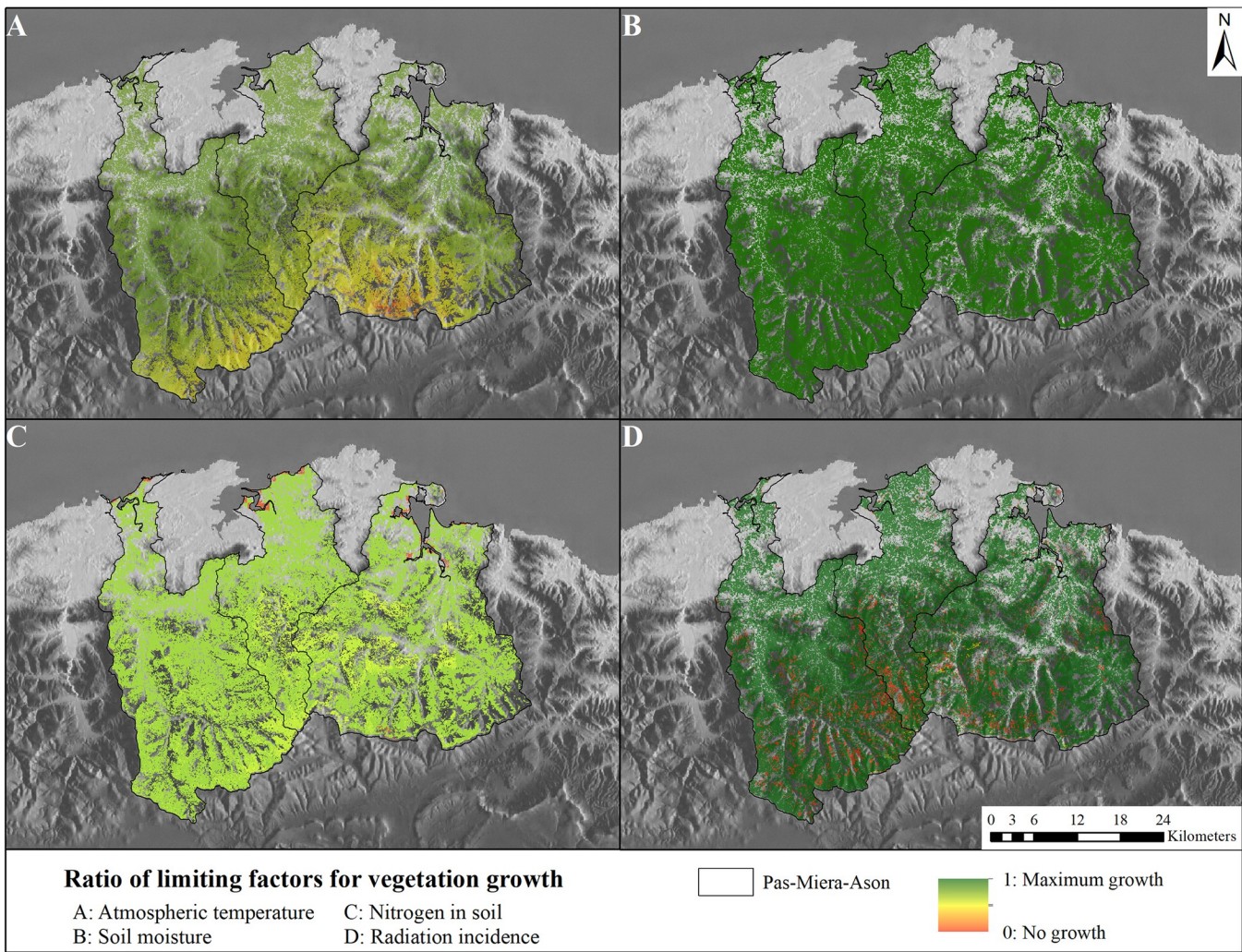

**Fig 6. Modelled results of vegetation limiting factors for May 2018.** All outputs range from 0 (no vegetation growth) to 1 (maximum vegetation growth). (A) Atmospheric temperature limitation. (B) Soil moisture limitation. (C) Soil nitrogen limitation. (D) Solar radiation incidence limitation. Reprinted from [67] under a CC BY license, with permission from ISPRS Journal of Photogrammetry and Remote Sensing, original copyright 2014.

**Table 3. Examples of parameters and values in the Puerto model.**

| IDMancha | com | com2 | t (count of timestep) | FT [0–1] | FR [0–1] | FH [0–1] | FN [0–1] |
|----------|-----|------|-----------------------|----------|----------|----------|----------|
| 442 | 9 | 9 | 1 | 0.701909 | 0.164 | 1 | 0.5 |
| 442 | 14 | 14 | 1 | 0.701909 | 0.164 | 1 | 0.65 |
| 442 | 28 | 28 | 1 | 0.701909 | 0.164 | 1 | 0.65 |
| 458 | 7 | 7 | 1 | 0.701909 | 0.116 | 1 | 0.65 |
| 458 | 13 | 13 | 1 | 0.701909 | 0.116 | 1 | 0.5 |

First lines of the "Vegetation Limiting Factor" internal table (same PaL model shown in Fig 6), accessible only for R software expert. The table indicates:

• IDMancha: code of observed plot

• com: code of the main vegetation

• com2: overstory vegetation, in case there is one

• t: timeline with daily timestep, always starting at the first of January of the year determined by the modeller; for example, "t = 1 means the first day in the dynamic Puerto model

• FT: mean parameter corresponding to temperature as vegetation limiting factor for each observed plot ("IDMancha), vegetation type ("com", "com2") and time ("t"). These parameters are between 0 and 1.

• FR: mean parameter corresponding to radiation as vegetation limiting factor for each observed plot ("IDMancha), vegetation type ("com", "com2") and time ("t"). These parameters are between 0 and 1.

• FH: mean parameter corresponding to Temperature as vegetation limiting factor for each observed plot ("IDMancha), vegetation type ("com", "com2") and time ("t"). These parameters are between 0 and 1.

• FN: mean parameter corresponding to nitrogen as vegetation limiting factor for each observed plot ("IDMancha), vegetation type ("com", "com2") and time ("t"). These parameters are between 0 and 1.

ranges. These effects are complex: soil characteristics affect water content, aspect affects shade patterns and the incidence of radiation, and elevation affects the temperatures and precipitation levels to which plants are exposed.

Fig 6 shows the influence of each variable managed vegetation growth in May 2018. While soil moisture (Fig 6B) and solar radiation incidence (Fig 6D) positively affect vegetation growth (except in some shaded areas in the case of solar radiation incidence), temperature (Fig 6A) has an increasing influence with elevation and nitrogen is the most uniformly limiting factor (Fig 6C). Puerto does not provide these results (Fig 6) in spatial form. An expert in R can extract the R internal table (Table 3), which contains outputs of the limiting factor for climate.

The advanced modellers can link the plot identification code to a vector dataset to spatialize the output. However, the distribution of vegetation within each plot is not given.

## Vegetation

The entire vegetation life cycle—including growth (S5 Fig and S5A-S5C Table in S1 File), senescence (S6 Fig and S6A-S6C Table in S1 File), and litterfall (S7 Fig and S7A-S7C Table in S1 File) —is composed of three different namespaces which include 44 component models. Vegetation life cycle is affected not only by climate, but also by livestock activity, nutrient uptake and human intervention, in particular by harvesting or fertilization cycles. With PaL, we can estimate the evolution of the parameters in each grid cell over time, depending on the type of vegetation. This group of models can be run with or without human and animal influence.

Fig 7A shows the potential vegetation growth under climatic factors (temperature, solar radiation and soil moisture). The results of Fig 7B are the actual growth model, based on potential growth but also taking into account nitrogen limitation and the influence of livestock on the grazing areas. Two notable trends emerge–first, that maximum potential daily vegetation growth is 5.29 grams per day, while actual growth is 1.58 grams per day. Second, the distribution of vegetation growth is heterogeneous, decreasing in mountainous areas than flatter areas (Fig 7B).

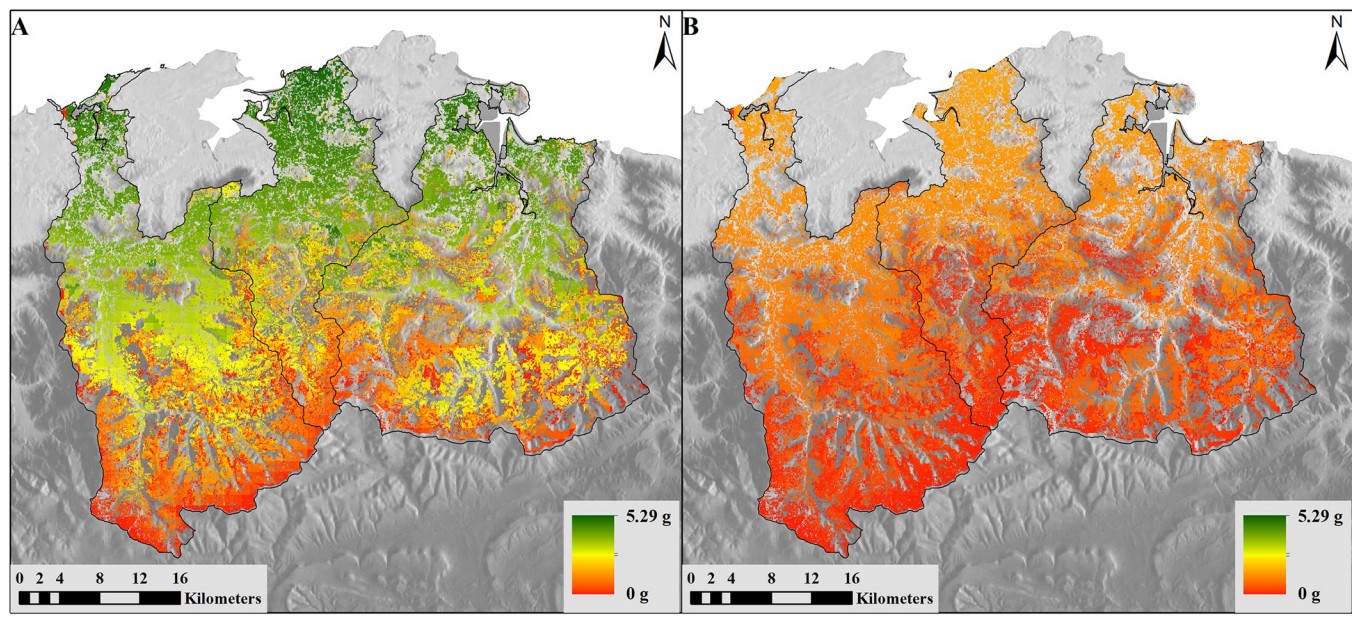

**Fig 7. Modelled results of vegetation growth.** (A) Potential Growth causing AboveGround Biomass model in grams/day and (B) Growth causing AboveGround Biomass model in grams/day (Cantabrian case study, May 2018). Reprinted from [67] under a CC BY license, with permission from ISPRS Journal of Photogrammetry and Remote Sensing, original copyright 2014.

Puerto vegetation growth outputs include tables in R or graphical bar and line graph outputs (Fig 8). The information on monthly average vegetation growth (bars) and livestock ingestion (lines) are shown for a period of years determined by the modeller, in this case, 5 years. Results are aspatial, as compared to the spatially explicit outputs for a flexible, user-defined time period in PaL. In addition, the legend of results is given only in Spanish. A translation to English is provided in the figure description.

## Livestock

The namespaces related to ingestion (S8 Fig and S8A-S8C Table in S1 File), excretion (S9 Fig and S9A-S9C Table in S1 File) and mass (S10 Fig and S10A-S10C Table in S1 File) of livestock include a total of 55 component models, including both cattle and mares. Key outputs include sustainability of the exploitation of pastures, biomass intake, the variation of livestock weight and the amount of excrement returned to the environment. Based on modelled livestock mass variation for cattle (Fig 9A) and mares (Fig 9B), cattle are more affected by topographic conditions and vegetation availability than mares.

Results depend not only on vegetation type and life cycle, but also on the estimated number of animals on each hectare of land, competition between them, accessibility to the vegetation, and topography, among other influences. The livestock weight can vary greatly during a year, at times showing negative change (Fig 10). The observation of weight variation can help managers to track the livestock condition and to know if the number of livestock and the forage production (carrying capacity) are in balance.

## Nitrogen cycle

The nitrogen cycle namespace includes all the models related to nitrogen in its different states and forms. The calculation of the nitrogen content in senesced leaves, mineral nitrogen present in the soil, nitrogen in livestock excrement and that used for plants are some of the models

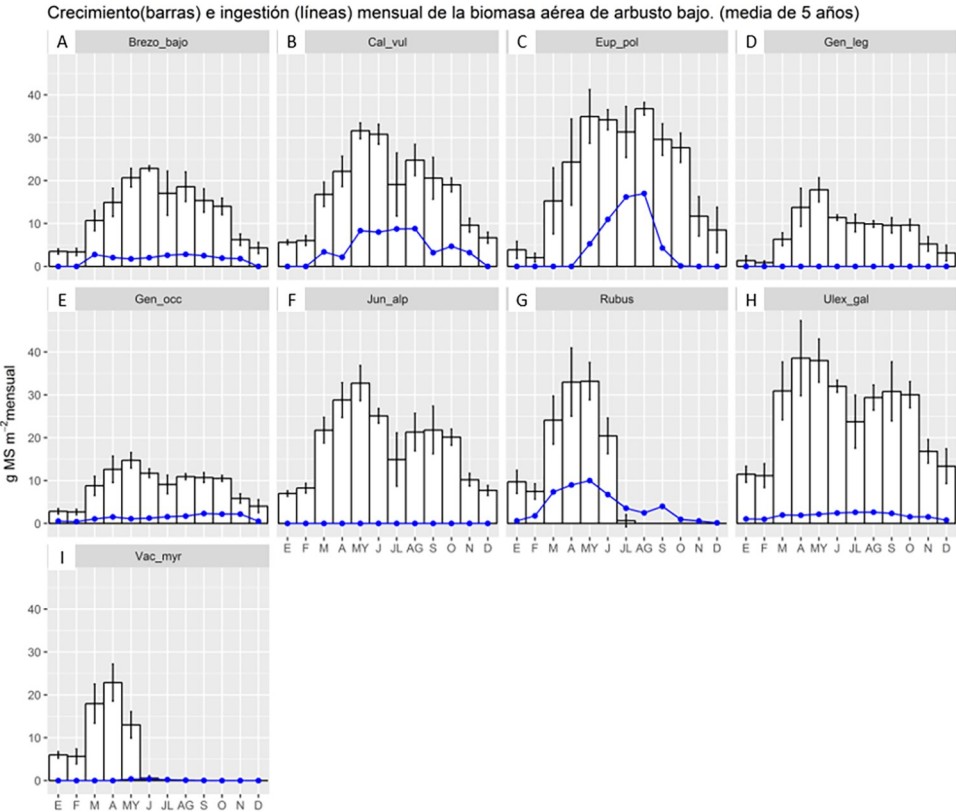

**Fig 8. Example of Puerto output.** Puerto can display results as images produced in R software (in Spanish language by default); this example shows the 5-yearly mean of scrubland vegetation growth (blue line) and livestock vegetation ingested (white bar) by month and type of vegetation. General title: Monthly growth (bar) and ingestion (line) of scrubland vegetation (5-year mean). Graph title: Abbreviated scientific name of scrub species. A: *Erica Vagans*, B: *Calluna vulgaris*, C: *Euphorbia Polygalifolia*, D: *Genista legionensis*, E: *Genista occidentalis*, F: *Juniperus alpina*, G: *Rubus ulmifolius*, H: *Ulex gallii*, I: *Vaccinium myrtillus*. X-axis: Grams of solid material by square meters and month. Y-axis: Month from January to December. White bar: Scrubland vegetation growth. Vertical line in each bar indicates the range between maximum and minimum scrubland vegetation growth during the 5 year by month. Blue line: Livestock vegetation ingested.

called on by this namespace. An interesting part of this namespace is the "Nitrogen leaching" model (Fig 11), which can interact with the models related to the water cycle within k.LAB for future studies of water quality and pasture management. The output of Puerto is an internal R table as Table 3.

## End-user features

**Output maps.** The first set of outputs provided to the end-user is a series of temporally explicit maps. Temporally dynamic outputs can be viewed using the "play" button at the bottom of the menu on the left side of Fig 12. A user can also view all the models computed as dependencies of the requested model. All results (main model and dependent models) can be downloaded in Geotiff format or as an image. In addition, basic information is provided such as total grid size, cell size, temporality, total observed model area, symbology and colour ramp style with labelling and a histogram for each of the model's inputs and outputs (Fig 13).

**Data flow.** k.LAB creates an interactive data flow of the requested model that is built on the fly (Fig 14). Thus, all the models and dependencies are shown. By clicking on each block of the data flow, more information is provided describing:

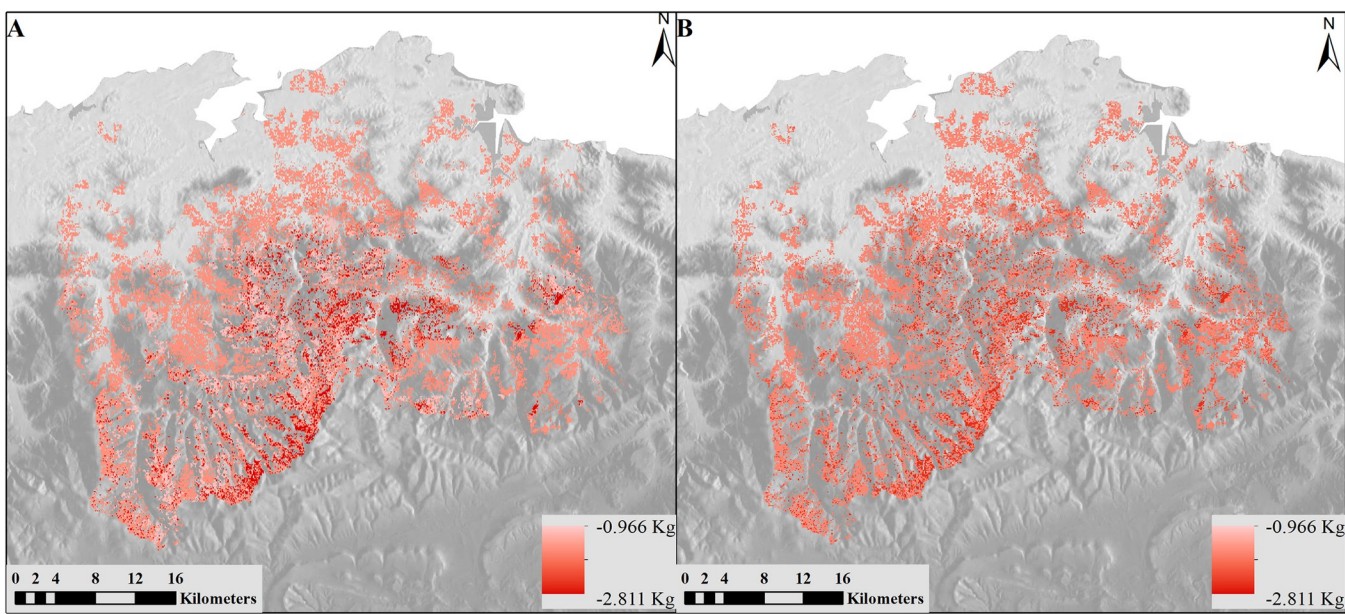

**Fig 9. Modelled results of livestock mass variation.** (A) Cattle and (B) Mares in kg/day. Reprinted from [67] under a CC BY license, with permission from ISPRS Journal of Photogrammetry and Remote Sensing, original copyright 2014.

1. for resources (data sources), basic information about the data source. This is based on meta-data contributed by users who have previously contributed data resources to the k.LAB network, including links back to the original data source; Fig 15A);

2. for tables, each table's composition (Fig 15B); and

3. for parameterised models, the expression or algorithm used (Fig 15C).

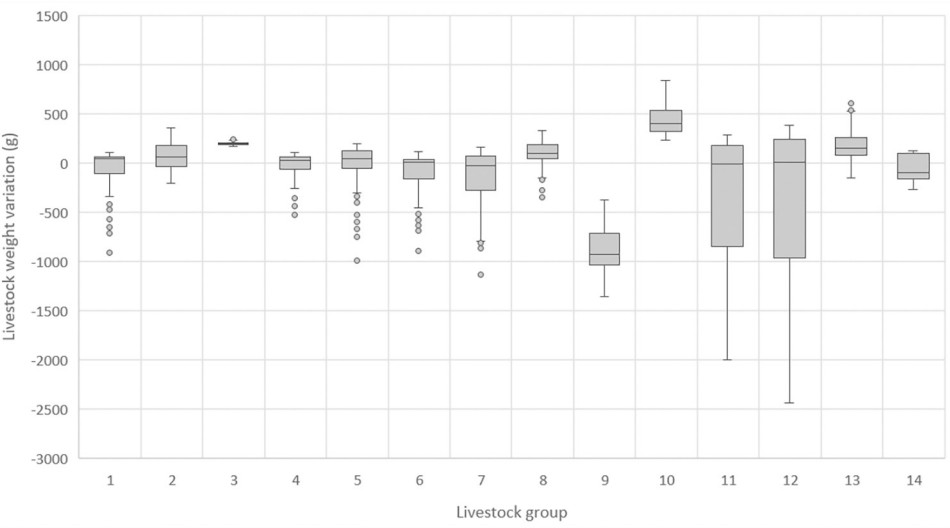

**Fig 10. Weight variation of 14 livestock groups in different plots during one year.**

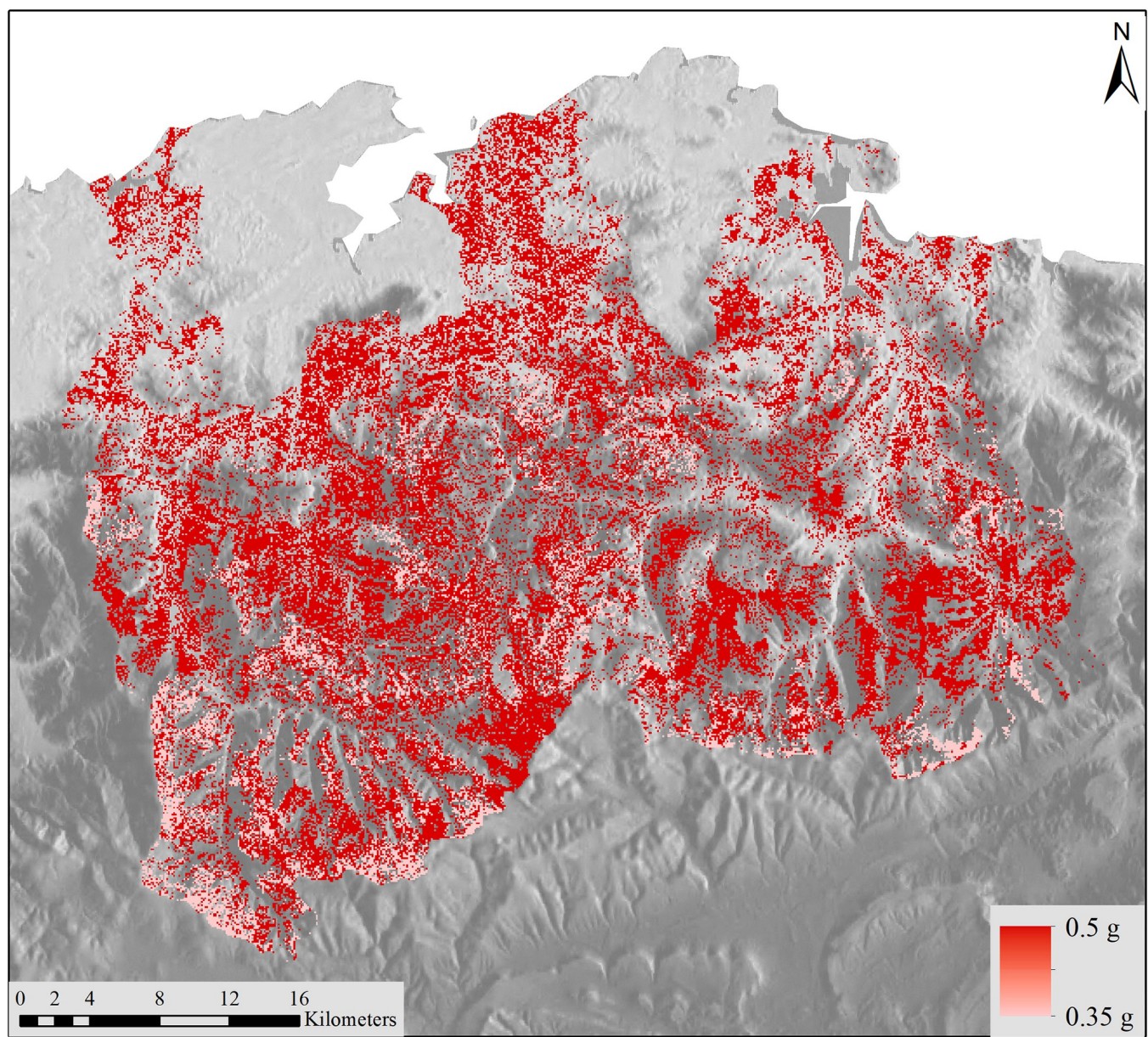

**Fig 11. Nitrogen leaching output model in grams of nitrogen mass.** Reprinted from [67] under a CC BY license, with permission from ISPRS Journal of Photogrammetry and Remote Sensing, original copyright 2014.

## Report

A printable report (Fig 16) is also created on the fly, collecting documentation from each model being run and adapting it to the results being calculated. Basic documentation about each model component is entered by each model's contributor in k.LAB, which is called when the model is run and assembled into the report; the modellers' documentation uses a template language that makes it possible to "react" to the results. This reporting facility complements the dataflow graph in making the system transparent and reliable. The report follows the standard structure of a scientific article (introduction, methods, results, discussion, conclusion and references). It can include tables, figures or other elements, depending on the model, and can be downloaded in.pdf format.

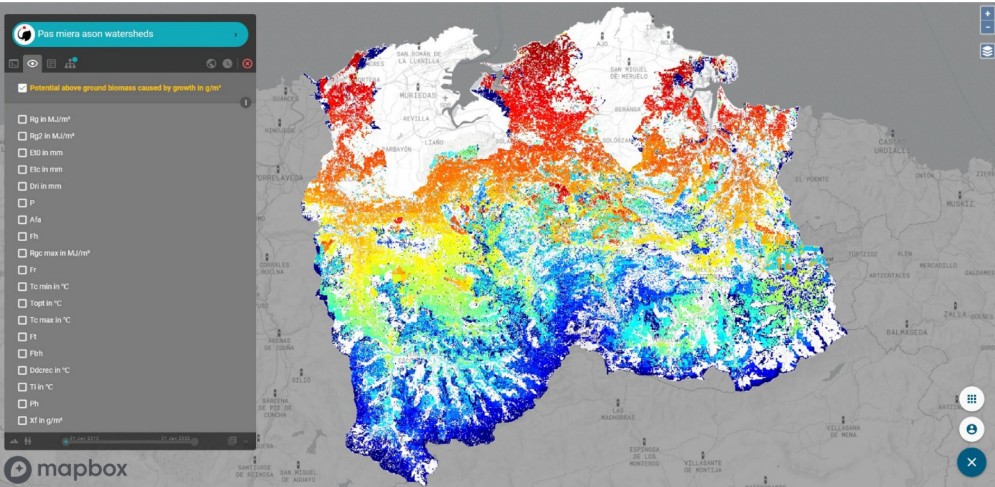

**Fig 12. Output of the model "Potential above ground biomass caused by growth" and its model dependencies.**
Reprinted from OpenLayers (https://openlayers.org) under a 2-Clause BSD, with permission from OpenLayers,
original copyright 2005 to present.

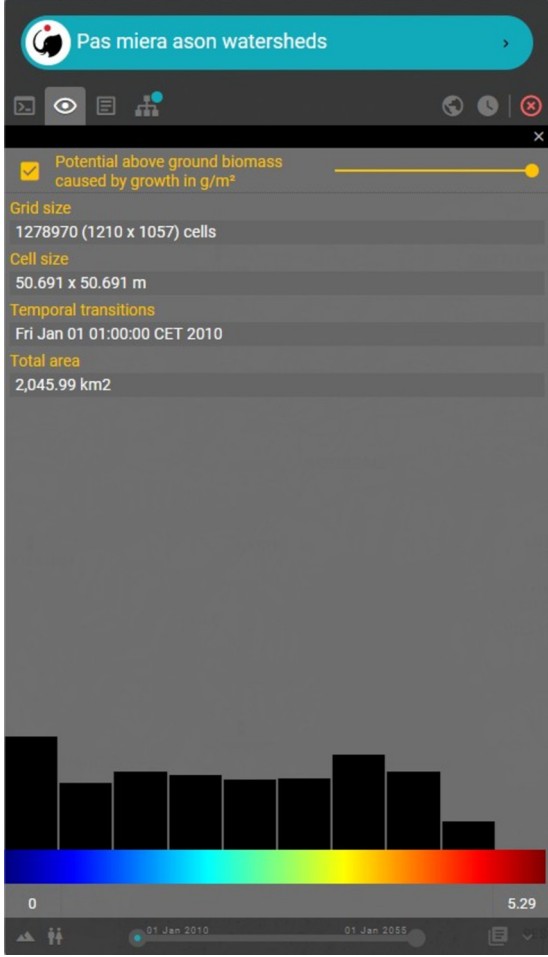

**Fig 13. Information related to each model output or dependency.**

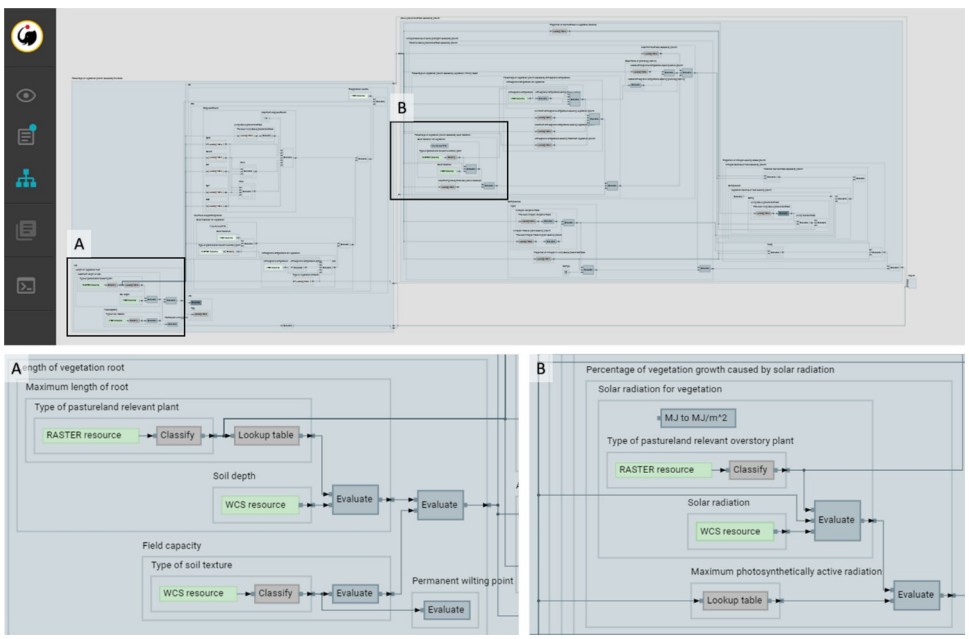

**Fig 14. Dataflow of "Above ground biomass caused by growth" model created on the fly as the model runs.** (A) and (B) show dataflow in detail and its boxes.

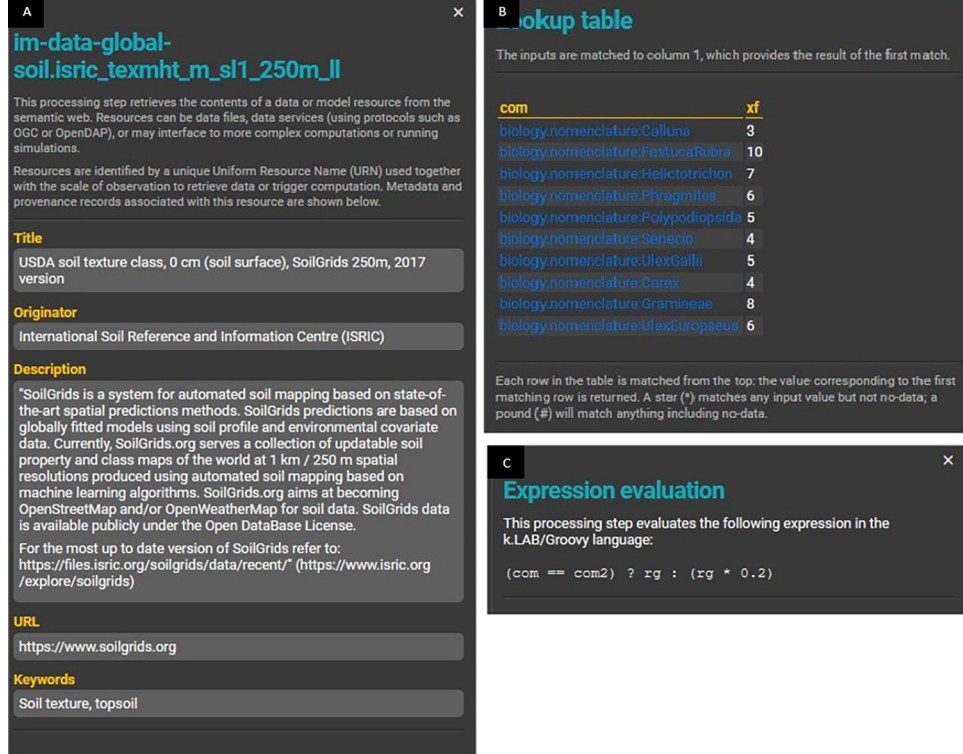

**Fig 15. Example of additional dataflow component views.** The individual dataflow components are shown by clicking on the boxes. (A) Information related to resources (i.e., raster data, vectorial), (B) information related to a look-up table where each ontology is linked to the identifier of the resource, (C) information related to a model equation. In this case, it is a conditional expression that indicates: when main vegetation (com) is equal to secondary vegetation (com2), then the result is the solar radiation (rg), otherwise, the result is rg multiplied by 0.2.

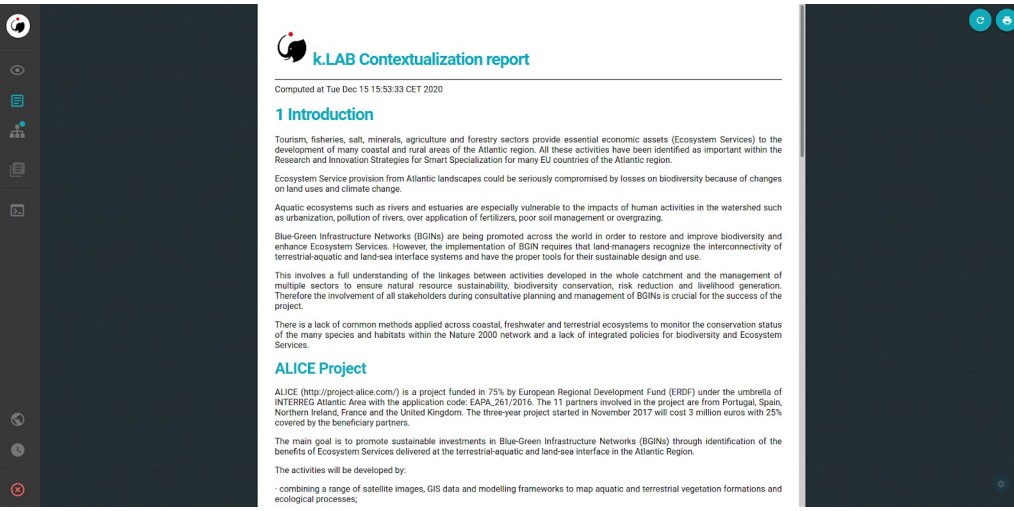

**Fig 16. Report describing the model, created on the fly as the model runs.**

## Discussion

A sustainable balance between agricultural production and healthy ecosystems in agricultural landscapes has been challenging to achieve. The main difficulties can be linked to population growth and people's increased demands for food, water and energy, the limited area of arable land to expand food production and increasing pressure on natural resources from various human activities [26, 74]. These factors are further compounded by land degradation and water contamination, climate change, sub-optimal agricultural and land-use policies and market fluctuations [75, 76]. The PaL models developed in k.LAB can be used to improve the management of agricultural systems by:

- integrating all the components of agricultural systems modelling in one platform,

- simulating the effects of alternative resource use strategies,

- improving the efficiency of low-input and intensive agricultural systems, and

- improving accessibility and transparency of simulation models to stakeholders.

The divergence in the time scales between farmer choices and environmental goals is a substantial management challenge. While farmers often need or want to fulfil their financial and land management objectives in the short term (i.e., months and seasons in this and the following year), environmental goals may take much longer to be reached (potentially years to decades). The temporal flexibility in modelling plays a key role to quantify short- and long-term processes in both the agricultural system and the environment. As we show in this article, the k.LAB approach ensures semantic consistency in temporal data, from historical observations to future scenarios, to respond to these needs in different situations. Moreover, the PaL namespaces could be expanded to simulate environmental disturbances, disease spread, climatic change and simulated management plans to deal with such challenges, building on the existing PaL namespaces and without having to change any of them and its models. For example, providing a model for "change in X" is all it takes to make a previously static model of concept X dynamic, as the k.LAB engine will automatically insert it in the dataflow whenever the context is computed over multiple timesteps.

Because environmental modelling, including pasture and livestock simulation, tends to be driven by the need to address case-specific issues, data and model reuse recommendations are often unclearly defined. Moreover, the collected data are often not made available to other researchers; when they are placed in public repositories data are often findable and accessible but lag in their interoperability and reusability [77]. As a result, in the best case substantial manual GIS processing is required before a user can work with previously generated data; in the worst case data may be lost entirely after the modelling results are published. In this article, we demonstrate a semantics-first approach to harmonize data and models of livestock and pastureland, in order to make them interoperable [64]. Thus, PaL's modular approach allows models and data to be combined for specific purposes in one platform, making the simulation process more efficient by representing diverse pieces of knowledge in the same system, which is a common difficulty in agricultural modelling systems [78]. This is a significant improvement in dealing with the complex interdependencies between humans and nature in agricultural systems, where data come from different sources and knowledge domains as in the case study presented.

The models, algorithms, data sources, and results described in this article are accessible to non-technical users through a web browser application, k.Explorer–a substantial improvement from the previous edition of the Puerto model, which was only available to technical modellers proficient in the R programming language. As described in the "End-user features" section of the Results, this makes scientific information more easily understandable and accessible, bringing scientific research closer to society with greater transparency (Figs 12–16).

While this article focused on a specific case study, both thematically and in geographical scope, we argue that this case is representative of common practice in scientific modelling whereby the developed knowledge is very seldom reused beyond the model developers [66].

k.LAB is an open and collaborative technology aiming to expand and improve the availability of interoperable data and models across disciplines [71, 79]. This technology can be used to substantially improve agricultural data and models' accessibility, harmonize them in order to facilitate their wider reuse, improve their quality and consistency. PaL models are made available to both farmers and policy makers as an open, reusable and efficient toolbox. Modellers can contribute new data and models and the knowledge to ensure their appropriate reuse through a dedicated interface [64]—a collective effort to provide stakeholders with the needed tools to face the new challenges in agriculture systems [55, 80].

The versatility and flexibility of this approach encourages model reusability, which is particularly valuable to iteratively update assessments as newer or more reliable information becomes available. Data inputs made available in the k.LAB system can affect PaL modelling outputs and other ecosystem services models connected through semantics (Fig 5). Both inputs and outputs from the PaL models can be automatically reused at different temporal and spatial scales, ranging from local analysis to national scales. Moreover, k.LAB automatically negotiates measures and their units based on the spatial and temporal context and resolution of the analysis, so different models can use different units.

We also note three limitations and complexities for the benefit of future investigations. First, input data needed to run PaL outside the Cantabrian case study region are available on the k.LAB network but may not have the same quality or resolution due when relying on global data. This could affect the reliability of PaL outputs when run outside the Cantabrian region. Hence, we recommend further validation of model outputs in future applications. Second, the types of modelled pastureland vegetation and livestock are currently limited to certain classes (Fig 2). Third, some excessively complicated models [81] could be replaced by simpler ones. This would require more accessible cloud-hosted data, but would simultaneously decrease computational needs.

This article demonstrated how agricultural modelling can be made more transparent and accessible. In particular, we showed how to run and produce results from the Pasture and Livestock (PaL) namespaces in the k.LAB modelling platform, capitalizing on a semantics-first approach [63]. We applied this set of models to a case study in the Cantabrian region of Spain, where complex interactions among vegetation, livestock, and nitrogen need to be disentangled for improved agroecosystem management. Additional agricultural models can be incorporated and connected with the currently available PaL models in the future. Some of these models may expand on other ecological aspects, such as pest, weed and disease spread or carbon and phosphorus cycling, which are closely linked to nitrogen. Others might expand on the micro-economics of farm operations, taking into account the cost-efficiency of management activities given farmers' current economic status. Similarly, the existing models can incorporate new input data related to vegetation and livestock species. Moreover, further research could analyse the interactions between PaL models and other ecosystem service models, to fully capture the complex implications of pasture management patterns [11, 65, 71].

## Conclusions

The evolution of agriculture and the challenges it faces, both in terms of productivity and ecological impacts, require focused efforts to design more sustainable agricultural systems. The case study in Cantabria addresses a set of environmental and agricultural management changes over the past decades. The current pressure of tourism and the trend of farmland abandonment are risking the balance between nature and society in these systems. One of the main challenges of this study was to combine, using a unified yet highly flexible and accessible approach, the biophysical, technical and management knowledge needed to analyze the current conditions and explore future trends.

In this article, we break down the original monolithic Puerto model, developed for managing rangelands in the Cantabrian region of Spain, into ten Pasture and Livestock k.LAB namespaces, composed of 198 models. We applied these a fine temporal and spatial scale over the case study area, the Pas, Miera and Ason watersheds in Cantabria, responding to the needs for modelling their extensive agricultural systems. To do so, we first provided insights into current and past agricultural trends derived from literature and expert knowledge regarding to the Cantabrian agroecosystem situation. Next, we developed an open and semantic modelling application for pasture and livestock modelling in the k.LAB platform. This provides stakeholders with an accessible and user-friendly web-browser with that better bridges the gap between technical scientific modelling and land managers. Accessible and context-dependent models can provide solutions for different needs, such as those of i) policy-makers, who can better monitor landscape performance and health, ii) farmers, who can simulate alternative management strategies and potential risks to farming production and devise adaptation strategies, and iii) scientists, who can contribute to greater knowledge reuse and application to on-the-ground decision making. Further work will integrate an optimization module that can assess the pasture sustainability [77, 78] to further facilitate the land managers' decisions.

This article elaborated the importance of overall modelling strategy and design for interoperability and reusability, showing how to improve the ease of use of scientific models and their application to decision making. Within a collaborative modelling system like k.LAB, all models are enhanced through wider community testing, reuse, and application to different contexts. Through wider reuse, models can become increasingly realistic, reliable and useful. This approach is applicable for a wide range of environmental modelling problems, though it is especially suitable for agricultural systems, where underlying data are gathered from different

sources and domains, as it facilitates a transdisciplinary scientific approach to complex modelling and management problems.

## Supporting information

**S1 Fig. General information.** A) Namespace dataflow and B) Model's dataflow legend.
(TIF)

**S2 Fig. Dataflow of moisture namespace.**
(TIF)

**S3 Fig. Dataflow of radiation namespace.**
(TIF)

**S4 Fig. Dataflow of temperature namespace.**
(TIF)

**S5 Fig. Dataflow of vegetation growth namespace.**
(TIF)

**S6 Fig. Dataflow of the senescence namespace.**
(TIF)

**S7 Fig. Dataflow of litterfall namespace.**
(TIF)

**S8 Fig. Dataflow of ingestion namespace.**
(TIF)

**S9 Fig. Dataflow of excretion namespace.**
(TIF)

**S10 Fig. Dataflow of livestock mass namespace.**
(TIF)

**S11 Fig.** A. Dataflow of nitrogen cycle namespace. B. Dataflow of nitrogen cycle namespace.
(TIF)

**S12 Fig. Results of vegetation growth (g/m$^2$).** A) Puerto image and B) PaL map.
(TIF)

**S13 Fig. Box plot comparing distribution of vegetation growth results by plot for Puerto and PaL.**
(TIF)

**S14 Fig. Line graph wit PaL and Puerto vegetation growth results by plot.**
(TIF)

**S1 File.**
(DOCX)

## Acknowledgments

The authors would like to thank Joan Busqué who created and shared the original Puerto model and the team lead by José Barquín at the Hydrological Institute of Cantabria (IHC). Special thanks to Simone Langhans and Ken Bagstad who suggested revisions to the article. Robinson et al. (2014) for logistic support for EarthEnv-DEM90.

## Author Contributions

**Investigation:** Alba Marquez Torres.

**Methodology:** Alba Marquez Torres, Stefano Balbi, Ferdinando Villa.

**Software:** Ferdinando Villa.

**Supervision:** Stefano Balbi, Ferdinando Villa.

**Validation:** Stefano Balbi.

**Visualization:** Alba Marquez Torres.

**Writing – original draft:** Alba Marquez Torres.

**Writing – review & editing:** Stefano Balbi, Ferdinando Villa.

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
