## [Decision Letter · Decision Letter 0]

17 May 2022

PONE-D-22-04207Scientific modelling can be accessible, interoperable and user friendly: An example for pasture and livestock modellingPLOS ONE

Dear Dr. Marquez Torres,

Thank you for submitting your manuscript to PLOS ONE. After careful consideration, we feel that it has merit but does not fully meet PLOS ONE’s publication criteria as it currently stands. Therefore, we invite you to submit a revised version of the manuscript that addresses the points raised during the review process.

ACADEMIC EDITOR: According to the review process and results, my current attitude is that the manuscript is on the rim of rejection due to three major criticisms. 1. The manuscript is more like a technical report than an academic paper. Actually academic paper should have theoretical highlights to enlighten further advancements. 2. There are faults in presentation. 3. What is the target reader community?

We look forward to receiving your revised manuscript.

Kind regards,

Ning Cai, Ph.D.

Academic Editor

PLOS ONE

Journal Requirements:

“This research is supported by the Basque Government through the BERC 2018-2021 program, by the Ikertzaile Doktoreentzako Hobekuntzarako doktoretza-ondoko Programa, by FPI grant from Spanish Ministry of Economy and Competitiveness MINECO through BC3 María de Maeztu excellence accreditation MDM-2017-0714 and by the EU Interreg Atlantic Area Programme 2014–2020 (EAPA_261/2016 ALICE).”

“AMT:

- Spanish Ministry of Economy and Competitiveness MINECO (BC3 María de Maeztu excellence accreditation MDM-2017-0714). https://www.aei.gob.es/convocatorias/buscador-convocatorias/apoyo-centros-excelencia-severo-ochoa-unidades-excelencia-2

AMT, SB, FV:

- Interreg Atlantic Area Programme 2014-2020 (EAPA 261 - 2016 ALICE). https://www.atlanticarea.eu/
https://project-alice.com/es/alice-project/

- Basque Government (BERC 2018-2021 program) https://www.euskadi.eus/ayuda_subvencion/2017/berc/web01-tramite/es/

SB, FV:

- Ikertzaile Doktoreentzako Hobekuntzarako doktoretza-ondoko Programme. https://www.euskadi.eus/ayuda_subvencion/2021/posdoc-berriak-orokorra/web01-tramite/es/

5. We note that Figures 1, 2, 6, 7, 9, 10 and 11 in your submission contain map images which may be copyrighted. All PLOS content is published under the Creative Commons Attribution License (CC BY 4.0), which means that the manuscript, images, and Supporting Information files will be freely available online, and any third party is permitted to access, download, copy, distribute, and use these materials in any way, even commercially, with proper attribution. For these reasons, we cannot publish previously copyrighted maps or satellite images created using proprietary data, such as Google software (Google Maps, Street View, and Earth). For more information, see our copyright guidelines: http://journals.plos.org/plosone/s/licenses-and-copyright.

 a. You may seek permission from the original copyright holder of Figures 1, 2, 6, 7, 9, 10 and 11 to publish the content specifically under the CC BY 4.0 license.  

Additional Editor Comments:

According to the review process and results, my current attitude is that the manuscript is on the rim of rejection due to three major criticisms. 1. The manuscript is more like a technical report than an academic paper. Actually academic paper should have theoretical highlights to enlighten further advancements. 2. There are faults in presentation. 3. What is the target reader community?

Reviewers' comments:

Reviewer's Responses to Questions

**Comments to the Author**

1. Is the manuscript technically sound, and do the data support the conclusions?

Reviewer #1: Yes

Reviewer #2: Yes

Reviewer #3: Yes

2. Has the statistical analysis been performed appropriately and rigorously? 

Reviewer #1: Yes

Reviewer #2: Yes

Reviewer #3: N/A

3. Have the authors made all data underlying the findings in their manuscript fully available?

Reviewer #1: Yes

Reviewer #2: No

Reviewer #3: Yes

4. Is the manuscript presented in an intelligible fashion and written in standard English?

Reviewer #1: Yes

Reviewer #2: Yes

Reviewer #3: Yes

5. Review Comments to the Author

Reviewer #1: This paper focuses on accessible, interoperable, and user-friendly scientific modeling. Its work is very valuable. However, some comments and suggestions that I hope would help perfecting the manuscript. My comments and suggestions are as follows.

1.Abstract, “The original model, named Puerto, is developed in the R language and includes 1,491 lines of code divided into 13 script files and linked to 19 input tables.” the sentence does not reflect the existing problem that arise, the significance of the study should be highlighted.

2.This study mainly shows the advantages of spatial visualization of the PaL model output. Is the reliability of the output results verified? for example, compare the reliability of PaL versus Puerto output. After all, the accuracy of the model is the primary purpose of the analysis.

3. Why was May chosen for modeling?

4.Fig. 9, I tried but really cannot understand why livestock mass variation of is negative?

5. The Pal model takes into account the vegetation growth cycle, changes in livestock quality, nitrogen cycle, etc. However, I don't know how to achieve the best economic optimal benefits in the output results, which may be more desired by users, and more meaningful.

6. The reference format is very bad, please check each reference carefully.

Reviewer #2: This is a work that purports to link pasture and livestock modeling with GIS in a manner that would be both user-friendly and suitable for land policy making. The authors are advised to consider the following:

a) The whole work is more likely to be useful as a technical report than a scientific paper. Although it has undeniably been written in a scientific manner, its scope of readership is restricted to scientific officers of the geographical area it refers to or to people involved in making livestock modeling software more user-friendly. I am not sure it is possible to make more appealing to a wider readership, but it certainly has to.

b) This is largely due to the fact that it shows how render a cumbersome existing software named “Puerto" into a more friendly and effective one. One problem is that “Puerto” is by far and largely unknown to experts. And the only documentation offered here about it here are four references: two papers by Busque, one by Bedia et al and one by Marcos et al. Of these papers, three are in Spanish and one in English, but only as a conference paper. In short, the reader can not access this system, and if one does not speak Spanish, is unable not gain adequate knowledge about it.

c) This is also due to the rather weak presentation of this system (“Puerto") in the paper, so the reader can not appreciate (at least to the extent that the authors would like) the significance of their effort to improve it by creating PaL etc. And, in any case, the paper gives the impression that the conversion of “Puerto" to "PaL" and its equivalents is of local significance only (for North-Western Spain).

d) Consequently, converting a local system to a new one so as to become more useful or user-friendly is of very limited interest to the journal’s readership.

e) Figure 15 displays the funding source (“Alice") only and does not elucidate the reader in what has been done.

f) Figure 8: There should be a unified legend explaining what is displayed by the figure. The legend is split in two: one in English and one in Spanish above the figure.

g) Table 3: It is unclear what is t=1 all throughout the table (what does t stand for?) and if it takes the same value everywhere then why bother about it?

h) Figures 13 and 14 are not explained adequately in the legend, particularly the window c at the down right part of the figure.

i) The title of the paper needs to change so as to correspond to the content of the paper: this is a local study aiming at improving a local software system so as to become more efficient and user-friendly. This does not justify the use of a general title like the one that has been used. And, certainly, since this is a local study, it should be indicated in the title also: the word “Spain” should appear somewhere (or any other indicating the geographical location that the study concerns).

j) The references might need some updating, as the most recent one is from 2020, and we are already running the year 2022.

Reviewer #3: The article presents an adaptation of the existing model Puerto to an improved version that is more accessible, interoperable and user friendly, and could thus be of greater use for agriculture stakeholders in the face of future challenges for the sector. The article is appropriate for publication in Plos One, however I do have a list of minor comments and suggestions on the main manuscript and the supporting information mostly aiming to improve clarity.

There was no page or line numbering on the manuscript which makes it difficult to comment. Below I’m using the page number of the submitted document and the related paragraph.

P9, par1: ‘one main issues’-> of the

P9, par2: ‘growth, senescence death and litterfall’-> death?

P10, par1: you are mentioning novel approaches, yet the most recent citation is almost a decade old. Try to incorporate recent advances in the field on this and/or related domains. This is also something that is missing from the discussion.

P13, par2: remove ‘with’

P13, par2: Please check syntax of the last sentence.

P18, par4: three main models are stated but four are described (moisture, radiation, temperature, nitrogen). Also, how are extreme events defined and quantified?

Figure 6. It’s not clear how these values are computed. Are the default values maintained and in each graph a different limitation is applied? At what values of the limiting factor and what are the units?

Figure 7 You point out differences between maximum potential and maximum actual growth. This could be better visualized if the same color range was applied on both graphs. If the range was 0-5.29g on both graphs then we’d see a shift of B graph towards yellow/red.

Figure 8: Caption needs to be more informative and/or translate the Spanish title (and axes). Lines are ingestion and bars vegetation? What do the error bars stand for?

Figure 9: Consider the same color range suggestion as figure 7. Also, why is the livestock mass decreasing? Maybe that’s worth commenting on.

Table 3: please inset units for the parameters.

Figure 13: A description of what A and B is should be provided in the caption

Supporting information

I’ve found the units for many of the described variables either inconsistent, missing or confusing. Please carefully go through the whole list and check for inconsistencies and complete missing information. if the same type of variables have indeed different units please clarify why. Some examples

- Proportions: in same cases the range ([0 1]) is given (eg adic, st), some are unitless (eg fhs, cirhift02) and some have units day (eg pmin, pimx).

- Mass: shouldn’t all variables related to contents, biomass, mass be in kg (eg ihcift)? Similar for concentrations (eg nogm2ift)?

- Temperature: some missing (eg st1)

- Shouldn’t digestibility be in % (eg digm, digv)?

- Table S9. Almost all units are missing.

- For pv weight is given in kg but right below for pvr it has no units.

- Table 10. What is MJ/t (eg emi), is it different to MJ/d (eg nmd2)?

- Energy in some cases it is given, in some omitted (eg enigan, enimd) and in same it is confusing (eg in egkg, if it’s the energy needed to increase 1kg of body weight then the unit should be simply MJ).

- Since you are using m^2, and m^3, then volume variables (eg dri, et0) should also be mm^3, not mm

Also please check numbering on figure S10

6. PLOS authors have the option to publish the peer review history of their article (what does this mean?). If published, this will include your full peer review and any attached files.

Reviewer #1: No

Reviewer #2: No

Reviewer #3: No

---

## [Author Response · Author response to Decision Letter 0]

20 Jul 2022

Additional Editor Comments:

According to the review process and results, my current attitude is that the manuscript is on the rim of rejection due to three major criticisms. 1. The manuscript is more like a technical report than an academic paper. Actually academic paper should have theoretical highlights to enlighten further advancements. 2. There are faults in presentation. 3. What is the target reader community?

We thank the editor and the reviewers for their time and suggestions. We address the 3 points in our responses to reviewers’ comments and below.

1. With this article we have strived not to simply report on a specific agricultural model, but to highlight more general and more scientifically relevant technical problems related to reusability and FAIRness in models. This a key issue especially in this period of history, in which we are living a crisis of scientific reproducibility, recognized in many publications, including some in PLoS.

• Sandve, G. K., Nekrutenko, A., Taylor, J. & Hovig, E. Ten Simple Rules for Reproducible Computational Research. PLoS Comput. Biol. 9, e1003285 (2013). 

• González-Beltrán, A. et al. From Peer-Reviewed to Peer-Reproduced in Scholarly Publishing: The Complementary Roles of Data Models and Workflows in Bioinformatics. PLoS ONE 10, e0127612 (2015).

2. For this reason, our paper spends quite a bit of effort illustrating the semantic integration technique used to transform a monolithic and hard-to-use model into a modular, readable and easy to validate formalism that has the potential of changing agricultural and socio-ecological modelling much beyond the specific model discussed. Getting the overall aim of our work through to specialist reviewers has been difficult, no doubt because of our own shortcoming in making it as clear as possible. With this revision, we trust we have corrected those faults. At the same time, we have made all possible efforts to make the original Puerto model more accessible and documented, as per comments of one reviewer. We also hope that the English-language description of Puerto improves transparency, not without noting a subtle, but clearly visible, bias against science published in non-English literature.

3. Specifically, our effort to clarify the target community and overall scope of the work didn’t come out straightforwardly for one reviewer. Originally, the target community was implied in the last lines of the original abstract:

“We argue that this work demonstrates key steps needed to create more Findable, Accessible, Interoperable and Reusable (FAIR) models. This is particularly essential in environments as complex as agricultural systems, where multidisciplinary knowledge needs to be integrated across diverse spatial and temporal scales in order to understand complex and changing problems.”

In the revision, the introduction provides added clarity and purpose: “The aim is to showcase to the “social-ecological” modelling community the delivery of an existing, monolithic model, into a more modular, transparent and accessible approach to potential end users, regional managers, farmers and other stakeholders.”

 

Comments to the Author

Reviewer #1: This paper focuses on accessible, interoperable, and user-friendly scientific modeling. Its work is very valuable. However, some comments and suggestions that I hope would help perfecting the manuscript. My comments and suggestions are as follows.

1. Abstract, “The original model, named Puerto, is developed in the R language and includes 1,491 lines of code divided into 13 script files and linked to 19 input tables.” the sentence does not reflect the existing problem that arise, the significance of the study should be highlighted.

We thank the Reviewer for their time and suggestion. We made the suggested changes in the manuscript (lines: 22-24 in the manuscript and 25-29 in the revised manuscript with trach changes).

2. This study mainly shows the advantages of spatial visualization of the PaL model output. Is the reliability of the output results verified? for example, compare the reliability of PaL versus Puerto output. After all, the accuracy of the model is the primary purpose of the analysis.

Indeed, spatialization is a significant advantage of the approach described in the article, which also makes the results not directly comparable for accuracy. However, to respond to your very good point, we have run a comparison exercise in Supplementary Information Appendix 2. For this we focus on the PaL vegetation growth model in the area that Puerto uses as teste case and that is provided in an online repository with the code (https://doi.org/10.5281/zenodo.6786419). Fig 1 shows the Puerto example location compared to our case study. 

Fig 1. Location of PaL case study and the area that Puerto has as example in their code. 

3. Why was May chosen for modelling?

We decided to initialize the model in May 2018 because it was the starting date of the vegetation distribution map, according to our data availability. Other resources that we used in PaL models are related to weather, topography or livestock distribution and did not have the same limitations (lines: 297-302 in the manuscript and 320-325 in the revised manuscript with trach changes).

4. Fig. 9, I tried but really cannot understand why livestock mass variation of is negative? 

At the date that the model was generated, the estimation of pasture availability (Supporting Information SA1, S5) was lower than the estimation for the energy needs of the livestock present in the plot (Supporting Information SA1, S10), that is why they have a decrease in weight. Energy needs are related to metabolism, the type of terrain they are on and the animal condition, among other things. In addition, it must be taken into account that there are cows and mares in the same field, increasing the competition for food. It is a useful tool that can be used by the farmer to know if the number of livestock and the forage production (carrying capacity) are balanced. Lines: 393-400 in the manuscript and 428-435 in the revised manuscript with trach changes.

For example, the following graph shows the variation of weight in different cattle livestock groups during one year. We also decided to add this figure to the article as it may help to better understand this point.

5. The Pal model takes into account the vegetation growth cycle, changes in livestock quality, nitrogen cycle, etc. However, I don't know how to achieve the best economic optimal benefits in the output results, which may be more desired by users, and more meaningful.

We haven’t included yet an optimization algorithm to respond to your questions and we regard this work as an important future development. The modular semantic environment indeed provides a way to include this module without any modification to the overall model. Nevertheless, an end user can decide how to manage their pastures depending on the results of the model. For example, the land manager can explore if the livestock is losing weight, if the vegetation is growing or if he needs to add more fertilizer or cut the grass, among others. 

In the future, we expect to add a model that calculates whether the pasture is sustainable or not, based on the carrying capacity, fertilizer or type of vegetation. This model would be based on scientific studies: 

• Holechek J, Pieper RD, Herbel CH. Range Management: Principles and Practices. 6th ed. Pearson Education; 2011. 456 p. 

• Elgersma A, Struik PC, Maesen LJG van der. Grassland science in perspective. [Internet]. Wageningen Agricultural University; 1996. Available from: https://research.wur.nl/en/publications/grassland-science-in-perspective

We have included this in the article as future developments in lines: 569-571 in the manuscript and 611-613 in the revised manuscript with trach changes.

6. The reference format is very bad, please check each reference carefully.

We made the suggested changes in the manuscript (lines: 586-831 in the manuscript and 633-878 in the revised manuscript with trach changes).

Reviewer #2: This is a work that purports to link pasture and livestock modeling with GIS in a manner that would be both user-friendly and suitable for land policy making. The authors are advised to consider the following:

a) The whole work is more likely to be useful as a technical report than a scientific paper. Although it has undeniably been written in a scientific manner, its scope of readership is restricted to scientific officers of the geographical area it refers to or to people involved in making livestock modeling software more user-friendly. I am not sure it is possible to make more appealing to a wider readership, but it certainly has to.

We thank the reviewer for their time and suggestions, which we believe have improved the quality and relevance of the manuscript for a wider audience. We address the broader issue of the study’s scope and wider relevance in our responses to comments B, C, D, and below.

As explained above, we wrote this article to illustrate a more efficient way to write, deliver and share models, not merely to discuss a rewrite of an existing model. After clarifying this further in our revision, we regard PLoS ONE as an appropriate outlet due to its generalist readership, and its ability to publish even technical pieces. We appreciate that the reviewer’s comments made us clarify our target audience which perhaps was implicit in the abstract: now we added “The aim is to showcase to the “social-ecological” modelling community the delivery of an existing, monolithic model, into a more modular, transparent and accessible approach to potential end users, regional managers, farmers and other stakeholders.”

Please see our responses below, also covering the issue of geographical scope.

b) This is largely due to the fact that it shows how render a cumbersome existing software named “Puerto" into a more friendly and effective one. One problem is that “Puerto” is by far and largely unknown to experts. And the only documentation offered here about it here are four references: two papers by Busque, one by Bedia et al and one by Marcos et al. Of these papers, three are in Spanish and one in English, but only as a conference paper. In short, the reader can not access this system, and if one does not speak Spanish, is unable not gain adequate knowledge about it.

We agree that this specific example is a locally developed and used model, with most of its documentation available in non-English languages.

Even though it has been used in real life by practitioners in Cantabria, references 38 and 47-49), Puerto’s (and other models’) accessibility is one dimension of the problems this paper addresses. We have also clarified how Puerto is not inventing anything, but using equations which are well established in the scientific literature (references 39-46), which we now provide as supplementary information (Supporting Information, S1 Appendix). 

Additionally, regarding the geographical scope, it must be noted that the work described in this article has made it possible to apply the implemented model in the four case studies of the ALICE project, which covers the Atlantic space much beyond northern Spain (Portugal, France and Northern Ireland). 

At the same time Puerto is just an example that resonate with many scientific models written for research purposes. Most of them end up published in an article and then forgotten. Some are used with “local significance”. Very few become internationally recognized. This article is about making all scientific knowledge more usable. 

c) This is also due to the rather weak presentation of this system (“Puerto") in the paper, so the reader can not appreciate (at least to the extent that the authors would like) the significance of their effort to improve it by creating PaL etc. And, in any case, the paper gives the impression that the conversion of “Puerto" to "PaL" and its equivalents is of local significance only (for North-Western Spain).

We believe we have improved the presentation of Puerto and elaborated further on the issue of “non-local significance”. As described in the response to comment B above, we have more comprehensively described Puerto, its uses, and its relevance for modelling both within and beyond Spain.

d) Consequently, converting a local system to a new one so as to become more useful or user-friendly is of very limited interest to the journal’s readership.

By revising the paper as described above to clarify that its novel contribution is the conversion of Puerto to PaL within the globally used ARIES system, we highlight the general path toward making models interoperable and the value of that for scientific users, beyond the application of a particular model.

e) Figure 15 displays the funding source (“Alice") only and does not elucidate the reader in what has been done.

We made the suggested changes in the manuscript (lines: 111-115 in the manuscript and 119-123 in the revised manuscript with trach changes).

f) Figure 8: There should be a unified legend explaining what is displayed by the figure. The legend is split in two: one in English and one in Spanish above the figure.

We made the suggested changes in the manuscript (lines: 370-380 in the manuscript and 403-413 in the revised manuscript with trach changes).

g) Table 3: It is unclear what is t=1 all throughout the table (what does t stand for?) and if it takes the same value everywhere then why bother about it?

Table 3 is the first five lines of “Vegetation Limiting Factor” internal table of Puerto model in R. “t” is counting the daily time-step in the timeline meaning “t=1” the first day in the dynamic model. We made the changes in the manuscript (lines: 327-342 in the manuscript and 350-365 in the revised manuscript with trach changes).

h) Figures 13 and 14 are not explained adequately in the legend, particularly the window c at the down right part of the figure.

Figures 13 and 14 have changed to 14 and 15 respectively. We made the suggested changes into the manuscript (lines: 436-445 in the manuscript and 476-486 in the revised manuscript with trach changes).

i) The title of the paper needs to change so as to correspond to the content of the paper: this is a local study aiming at improving a local software system so as to become more efficient and user-friendly. This does not justify the use of a general title like the one that has been used. And, certainly, since this is a local study, it should be indicated in the title also: the word “Spain” should appear somewhere (or any other indicating the geographical location that the study concerns).

We already responded on the non-local significance of the case study. The article is about making scientific models (often developed on case by case demand, often from scratch), more available as modular components, to be integrated in other computational workflows and help build up collective knowledge about human-natural integrated systems.

While Puerto can be defined as a “local” software, K.LAB itself is definitely not a local software. ARIES (powered by k.LAB) is endorsed by the UN DESA for natural capital accounting, mostly for its approach to interoperability. More information available at https://seea.un.org/content/aries-for-seea or in this recent publication:

• Balbi, S., Bagstad, K.J., Magrach, A. et al. The global environmental agenda urgently needs a semantic web of knowledge. Environ Evid 11, 5 (2022). https://doi.org/10.1186/s13750-022-00258-y

We made sure to clarify the scope of this technology. Nevertheless we agree with you that the title is too generic, thus we proposed a new one: “Scientific modelling can be accessible, interoperable and user friendly: A case study for pasture and livestock modelling in Spain”.

j) The references might need some updating, as the most recent one is from 2020, and we are already running the year 2022.

We made the suggested changes into the manuscript (references: 57-60, 66).

Reviewer #3: The article presents an adaptation of the existing model Puerto to an improved version that is more accessible, interoperable and user friendly, and could thus be of greater use for agriculture stakeholders in the face of future challenges for the sector. The article is appropriate for publication in Plos One, however I do have a list of minor comments and suggestions on the main manuscript and the supporting information mostly aiming to improve clarity.

There was no page or line numbering on the manuscript which makes it difficult to comment. 

We thank the Reviewer for his/her time and suggestion. We added line numbering in the manuscript.

Below I’m using the page number of the submitted document and the related paragraph.

• P9, par1: ‘one main issues’-> of the

We made the suggested changes in the manuscript (lines: 71 in the manuscript and 78 in the revised manuscript with trach changes).

• P9, par2: ‘growth, senescence death and litterfall’-> death?

We made the changes in the manuscript (lines: 84 in the manuscript and 91 in the revised manuscript with trach changes).

• P10, par1: you are mentioning novel approaches, yet the most recent citation is almost a decade old. Try to incorporate recent advances in the field on this and/or related domains. This is also something that is missing from the discussion.

We made the suggested changes in the manuscript (references: 56-60).

• P13, par2: remove ‘with’

We made the suggested changes in the manuscript (lines: 169 in the manuscript and 179 in the revised manuscript with trach changes).

• P13, par2: Please check syntax of the last sentence.

We made the suggested changes in the manuscript (lines: 173-175 in the manuscript and 184-187 in the revised manuscript with trach changes).

• P18, par4: three main models are stated but four are described (moisture, radiation, temperature, nitrogen). Also, how are extreme events defined and quantified? 

We made the suggested changes in the manuscript (lines: 304 in the manuscript and 328 in the revised manuscript with trach changes). Extreme events are not quantified itself, but are implicit in some of the models. As weather data is an input of some models related to vegetation and, each type of vegetation has limiting vital parameters (minimum and maximum temperature, radiation and humidity to live), any event outside these vital parameters will influence the outcome of several models, as the vegetation will not be able to grow or will die. For example, some models that are influenced directly or indirectly are the vegetation growth model, vegetation death or livestock ingestion, among others.

• Figure 6. It’s not clear how these values are computed. Are the default values maintained and in each graph a different limitation is applied? At what values of the limiting factor and what are the units? 

The model values that limit the growth of vegetation are between 0 and 1. These values have no units as it is an index that is created after several calculations (Supporting Information S1 Appendix: S2-4 Fig and S2-4 A-C Table). It is calculated taking into account whether the environmental conditions affecting the vegetation (temperature, radiation, humidity and nitrogen) are sufficient for it to grow. If the vegetation is able to reach its growth needs through the inputs from the received environmental conditions, it will grow the maximum possible [Ratio of limiting factor for vegetation growth = 1], equalling the actual growth to the potential growth. If, on the other hand, environmental conditions are adverse, the "Ratio of limiting factor for vegetation growth" will be smaller than 1, reaching 0 when there is no growth. Lines: 315-318 in the manuscript and 338-341 in the revised manuscript with trach changes.

• Figure 7 You point out differences between maximum potential and maximum actual growth. This could be better visualized if the same color range was applied on both graphs. If the range was 0-5.29g on both graphs then we’d see a shift of B graph towards yellow/red.

We made the suggested changes in the manuscript (lines: 359-362 in the manuscript and 385-388 in the revised manuscript with trach changes).

• Figure 8: Caption needs to be more informative and/or translate the Spanish title (and axes). Lines are ingestion and bars vegetation? What do the error bars stand for?

We made the suggested changes in the manuscript (lines: 370-380 in the manuscript and 403-413 in the revised manuscript with trach changes).

• Figure 9: Consider the same color range suggestion as figure 7. Also, why is the livestock mass decreasing? Maybe that’s worth commenting on.

We change the color in the manuscript (lines: 388 in the manuscript and 423 in the revised manuscript with trach changes). The model has been generated for time 1, the livestock have not yet been able to eat enough to compensate for the energy they need, so the weight is negative. They lose more weight in areas with steeper slopes, where less vegetation grows and the effort to move is greater. Moreover, for this case study, in the majority of plots, cows and mares coexist, creating a greater competition between them.

• Table 3: please inset units for the parameters. 

The parameters in the Table 3 have no units. “IDMancha”, “com”, “com2” and “t” are qualitative (the numbers are codes related to the variables). The parameters “FT”, “FR”, “FH” and “FN” are rankings valued 0 to 1. We clarified this in the manuscript (lines: 327-342 in the manuscript and 350-365 in the revised manuscript with trach changes).

• Figure 13: A description of what A and B is should be provided in the caption

Figure 13 have changed to Figure 14. We made the suggested changes in the manuscript (lines: 435-438 in the manuscript and 476-479 in the revised manuscript with trach changes).

• Supporting information

o I’ve found the units for many of the described variables either inconsistent, missing or confusing. Please carefully go through the whole list and check for inconsistencies and complete missing information. if the same type of variables have indeed different units please clarify why. Some examples:

o Proportions: in same cases the range ([0 1]) is given (eg adic, st), some are unitless (eg fhs, cirhift02) and some have units day (eg pmin, pimx).

We made the changes in the Supporting Information S1 Appendix file.

o Mass: shouldn’t all variables related to contents, biomass, mass be in kg (eg ihcift)? Similar for concentrations (eg nogm2ift)? 

Puerto has different units for mass, volume or area depending on the variable and transform the units in the code accordingly. For example, in the line of R code from the "Port" model:

, where “VarPesoInd” (livestock weight change) is in grams and PVr (livestock weight) in in kg. In the case of PaL, this calculation is not necessary because k.LAB does the transformation automatically. Therefore, the script in k.IM language is:

o Temperature: some missing (eg st1)

We made the suggested changes in the Supporting Information S1 Appendix file.

o Shouldn’t digestibility be in % (eg digm, digv)? 

The values are in fact proportions, that is why we have put [0-1] in units instead of the symbol %.

o Table S9. Almost all units are missing.

We made the suggested changes in the Supporting Information S1 Appendix file.

o For pv weight is given in kg but right below for pvr it has no units.

We made the suggested changes in the Supporting Information S1 Appendix file.

o Table 10. What is MJ/t (eg emi), is it different to MJ/d (eg nmd2)? 

o The Puerto manual uses the symbol "t" in order to identify each time step in the model, which in this case it refers to day (in Puerto as "d"). At the time of rewriting and adapting part of the Port manual, we have made a typo error as those symbology is confusing and is not using SI conventions.Energy in some cases it is given, in some omitted (eg enigan, enimd) and in same it is confusing (eg in egkg, if it’s the energy needed to increase 1kg of body weight then the unit should be simply MJ).

We made the suggested changes in the Supporting Information S1 Appendix file. “egkg” means energy needed for increase one kilogram, for this reason the units are MJ/kg.

o Since you are using m^2, and m^3, then volume variables (eg dri, et0) should also be mm^3, not mm. 

Milimeters (mm) are describing a density of a volume over area (mm3/mm2). It is used in incident volumes representing water content on an area as, for example, precipitation over an area; but it cannot represent water content in the soil. In the case of water content in the soil, the unit m3/m3 is refers to a given volume of water contained in given volume of substrate, as the water content at field capacity or the water content at wilting point1. However, k.LAB can corrects and rescales units based on spatial and temporal context in case that there is no consistency between models. (lines: 521-523 in the manuscript and 562-564 in the revised manuscript with trach changes).

1FAO: Allen RG, Pereira LS, Raes D, Smith M. Crop evapotranspiration - Guidelines for computing crop water requirements - FAO Irrigation and drainage paper 56. FAO (Food and Agriculture Organization of the United Nation); 1998. Available from: http://www.fao.org/3/x0490e/x0490e00.htm#Contents; page 162, equation 82 and 83, where ADT is as TAW and AFA is as RAW.

o Also please check numbering on figure S10

We made the suggested changes in the Supporting Information S1 Appendix file.

---

## [Decision Letter · Decision Letter 1]

15 Aug 2022

PONE-D-22-04207R1

Scientific modelling can be accessible, interoperable and user friendly: A case study for pasture and livestock modelling in Spain

PLOS ONE

Dear Dr. Marquez Torres,

Thank you for submitting your manuscript to PLOS ONE. After careful consideration, we have decided that your manuscript does not meet our criteria for publication and must therefore be rejected.

I am sorry that we cannot be more positive on this occasion, but hope that you appreciate the reasons for this decision.

Kind regards,

Ning Cai, Ph.D.

Academic Editor

PLOS ONE

Additional Editor Comments:

One reviewer is still unsatisfied at the revision. Since the manuscript has undergone a rigorous revision, this means that the criticism is essential and could not be resolved by further revisions.

Reviewers' comments:

Reviewer's Responses to Questions

**Comments to the Author**

1. If the authors have adequately addressed your comments raised in a previous round of review and you feel that this manuscript is now acceptable for publication, you may indicate that here to bypass the “Comments to the Author” section, enter your conflict of interest statement in the “Confidential to Editor” section, and submit your "Accept" recommendation.

Reviewer #2: (No Response)

Reviewer #3: All comments have been addressed

2. Is the manuscript technically sound, and do the data support the conclusions?

Reviewer #2: Yes

Reviewer #3: Yes

3. Has the statistical analysis been performed appropriately and rigorously? 

Reviewer #2: Yes

Reviewer #3: N/A

4. Have the authors made all data underlying the findings in their manuscript fully available?

Reviewer #2: Yes

Reviewer #3: Yes

5. Is the manuscript presented in an intelligible fashion and written in standard English?

Reviewer #2: Yes

Reviewer #3: Yes

6. Review Comments to the Author

Reviewer #2: The paper still appears more like a technical report that concerns a local case study rather than a substantial contribution to science that would be beneficial to many scientists outside Spain.

Reviewer #3: The authors have sufficiently addressed my previous comments and the clarity of the manuscript has significantly improved.

7. PLOS authors have the option to publish the peer review history of their article (what does this mean?). If published, this will include your full peer review and any attached files.

Reviewer #2: No

Reviewer #3: No

- - - - -

---

## [Author Response · Author response to Decision Letter 1]

14 Sep 2022

Comments to the Author

Reviewer #1: This paper focuses on accessible, interoperable, and user-friendly scientific

modeling. Its work is very valuable. However, some comments and suggestions that I hope would

help perfecting the manuscript. My comments and suggestions are as follows.

1. Abstract, “The original model, named Puerto, is developed in the R language and includes 1,491

lines of code divided into 13 script files and linked to 19 input tables.” the sentence does not reflect

the existing problem that arise, the significance of the study should be highlighted.

We thank the Reviewer for their time and suggestion. We made the suggested changes in the

manuscript (lines: 22-24 in the manuscript and 25-29 in the revised manuscript with trach changes).

2. This study mainly shows the advantages of spatial visualization of the PaL model output. Is the

reliability of the output results verified? for example, compare the reliability of PaL versus Puerto

output. After all, the accuracy of the model is the primary purpose of the analysis.

Indeed, spatialization is a significant advantage of the approach described in the article, which also

makes the results not directly comparable for accuracy. However, to respond to your very good

point, we have run a comparison exercise in Supplementary Information Appendix 2. For this we

focus on the PaL vegetation growth model in the area that Puerto uses as teste case and that is

provided in an online repository with the code (https://doi.org/10.5281/zenodo.6786419). Fig 1

shows the Puerto example location compared to our case study.

3. Why was May chosen for modelling?

We decided to initialize the model in May 2018 because it was the starting date of the vegetation

distribution map, according to our data availability. Other resources that we used in PaL models are

related to weather, topography or livestock distribution and did not have the same limitations (lines:

297-302 in the manuscript and 320-325 in the revised manuscript with trach changes).

4. Fig. 9, I tried but really cannot understand why livestock mass variation of is negative?

At the date that the model was generated, the estimation of pasture availability (Supporting

Information SA1, S5) was lower than the estimation for the energy needs of the livestock present in

the plot (Supporting Information SA1, S10), that is why they have a decrease in weight. Energy

needs are related to metabolism, the type of terrain they are on and the animal condition, among

other things. In addition, it must be taken into account that there are cows and mares in the same

field, increasing the competition for food. It is a useful tool that can be used by the farmer to know

if the number of livestock and the forage production (carrying capacity) are balanced. Lines: 393-

400 in the manuscript and 428-435 in the revised manuscript with trach changes.

For example, the following graph shows the variation of weight in different cattle livestock groups

during one year. We also decided to add this figure to the article as it may help to better understand

this point.

5. The Pal model takes into account the vegetation growth cycle, changes in livestock quality,

nitrogen cycle, etc. However, I don't know how to achieve the best economic optimal benefits in

the output results, which may be more desired by users, and more meaningful.

We haven’t included yet an optimization algorithm to respond to your questions and we regard this

work as an important future development. The modular semantic environment indeed provides a

way to include this module without any modification to the overall model. Nevertheless, an end

user can decide how to manage their pastures depending on the results of the model. For example,

the land manager can explore if the livestock is losing weight, if the vegetation is growing or if he

needs to add more fertilizer or cut the grass, among others.

In the future, we expect to add a model that calculates whether the pasture is sustainable or not,

based on the carrying capacity, fertilizer or type of vegetation. This model would be based on

scientific studies:

• Holechek J, Pieper RD, Herbel CH. Range Management: Principles and Practices. 6th ed. Pearson

Education; 2011. 456 p.

• Elgersma A, Struik PC, Maesen LJG van der. Grassland science in perspective. [Internet].

Wageningen Agricultural University; 1996. Available from:

https://research.wur.nl/en/publications/grassland-science-in-perspective

We have included this in the article as future developments in lines: 569-571 in the manuscript and

611-613 in the revised manuscript with trach changes.

6. The reference format is very bad, please check each reference carefully.

We made the suggested changes in the manuscript (lines: 586-831 in the manuscript and 633-878 in

the revised manuscript with trach changes).

Reviewer #2: This is a work that purports to link pasture and livestock modeling with GIS in a

manner that would be both user-friendly and suitable for land policy making. The authors are

advised to consider the following:

a) The whole work is more likely to be useful as a technical report than a scientific paper. Although

it has undeniably been written in a scientific manner, its scope of readership is restricted to

scientific officers of the geographical area it refers to or to people involved in making livestock

modeling software more user-friendly. I am not sure it is possible to make more appealing to a

wider readership, but it certainly has to.

We thank the reviewer for their time and suggestions, which we believe have improved the quality

and relevance of the manuscript for a wider audience. We address the broader issue of the study’s

scope and wider relevance in our responses to comments B, C, D, and below.

As explained above, we wrote this article to illustrate a more efficient way to write, deliver and

share models, not merely to discuss a rewrite of an existing model. After clarifying this further in

our revision, we regard PLoS ONE as an appropriate outlet due to its generalist readership, and its

ability to publish even technical pieces. We appreciate that the reviewer’s comments made us

clarify our target audience which perhaps was implicit in the abstract: now we added “The aim is to

showcase to the “social-ecological” modelling community the delivery of an existing, monolithic

model, into a more modular, transparent and accessible approach to potential end users, regional

managers, farmers and other stakeholders.”

Please see our responses below, also covering the issue of geographical scope.

b) This is largely due to the fact that it shows how render a cumbersome existing software named

“Puerto" into a more friendly and effective one. One problem is that “Puerto” is by far and largely

unknown to experts. And the only documentation offered here about it here are four references: two

papers by Busque, one by Bedia et al and one by Marcos et al. Of these papers, three are in Spanish

and one in English, but only as a conference paper. In short, the reader can not access this system,

and if one does not speak Spanish, is unable not gain adequate knowledge about it.

We agree that this specific example is a locally developed and used model, with most of its

documentation available in non-English languages.

Even though it has been used in real life by practitioners in Cantabria, references 38 and 47-49),

Puerto’s (and other models’) accessibility is one dimension of the problems this paper addresses.

We have also clarified how Puerto is not inventing anything, but using equations which are well

established in the scientific literature (references 39-46), which we now provide as supplementary

information (Supporting Information, S1 Appendix).

Additionally, regarding the geographical scope, it must be noted that the work described in this

article has made it possible to apply the implemented model in the four case studies of the ALICE

project, which covers the Atlantic space much beyond northern Spain (Portugal, France and

Northern Ireland).

At the same time Puerto is just an example that resonate with many scientific models written for

research purposes. Most of them end up published in an article and then forgotten. Some are used

with “local significance”. Very few become internationally recognized. This article is about making

all scientific knowledge more usable.

c) This is also due to the rather weak presentation of this system (“Puerto") in the paper, so the

reader can not appreciate (at least to the extent that the authors would like) the significance of their

effort to improve it by creating PaL etc. And, in any case, the paper gives the impression that the

conversion of “Puerto" to "PaL" and its equivalents is of local significance only (for North-Western

Spain).

We believe we have improved the presentation of Puerto and elaborated further on the issue of

“non-local significance”. As described in the response to comment B above, we have more

comprehensively described Puerto, its uses, and its relevance for modelling both within and beyond

Spain.

d) Consequently, converting a local system to a new one so as to become more useful or user-

friendly is of very limited interest to the journal’s readership.

By revising the paper as described above to clarify that its novel contribution is the conversion of

Puerto to PaL within the globally used ARIES system, we highlight the general path toward making

models interoperable and the value of that for scientific users, beyond the application of a

particular model.

e) Figure 15 displays the funding source (“Alice") only and does not elucidate the reader in what

has been done.

We made the suggested changes in the manuscript (lines: 111-115 in the manuscript and 119-123 in

the revised manuscript with trach changes).

f) Figure 8: There should be a unified legend explaining what is displayed by the figure. The legend

is split in two: one in English and one in Spanish above the figure.

We made the suggested changes in the manuscript (lines: 370-380 in the manuscript and 403-413 in

the revised manuscript with trach changes).

g) Table 3: It is unclear what is t=1 all throughout the table (what does t stand for?) and if it takes

the same value everywhere then why bother about it?

Table 3 is the first five lines of “Vegetation Limiting Factor” internal table of Puerto model in R.

“t” is counting the daily time-step in the timeline meaning “t=1” the first day in the dynamic

model. We made the changes in the manuscript (lines: 327-342 in the manuscript and 350-365 in

the revised manuscript with trach changes).

h) Figures 13 and 14 are not explained adequately in the legend, particularly the window c at the

down right part of the figure.

Figures 13 and 14 have changed to 14 and 15 respectively. We made the suggested changes into the

manuscript (lines: 436-445 in the manuscript and 476-486 in the revised manuscript with trach

changes).

i) The title of the paper needs to change so as to correspond to the content of the paper: this is a

local study aiming at improving a local software system so as to become more efficient and user-

friendly. This does not justify the use of a general title like the one that has been used. And,

certainly, since this is a local study, it should be indicated in the title also: the word “Spain” should

appear somewhere (or any other indicating the geographical location that the study concerns).

We already responded on the non-local significance of the case study. The article is about making

scientific models (often developed on case by case demand, often from scratch), more available as

modular components, to be integrated in other computational workflows and help build up

collective knowledge about human-natural integrated systems.

While Puerto can be defined as a “local” software, K.LAB itself is definitely not a local software.

ARIES (powered by k.LAB) is endorsed by the UN DESA for natural capital accounting, mostly

for its approach to interoperability. More information available at https://seea.un.org/content/aries-

for-seea or in this recent publication:

• Balbi, S., Bagstad, K.J., Magrach, A. et al. The global environmental agenda urgently needs a

semantic web of knowledge. Environ Evid 11, 5 (2022). https://doi.org/10.1186/s13750-022-

00258-y

We made sure to clarify the scope of this technology. Nevertheless we agree with you that the title

is too generic, thus we proposed a new one: “Scientific modelling can be accessible, interoperable

and user friendly: A case study for pasture and livestock modelling in Spain”.

j) The references might need some updating, as the most recent one is from 2020, and we are

already running the year 2022.

We made the suggested changes into the manuscript (references: 57-60, 66).

Reviewer #3: The article presents an adaptation of the existing model Puerto to an improved

version that is more accessible, interoperable and user friendly, and could thus be of greater use for

agriculture stakeholders in the face of future challenges for the sector. The article is appropriate for

publication in Plos One, however I do have a list of minor comments and suggestions on the main

manuscript and the supporting information mostly aiming to improve clarity.

There was no page or line numbering on the manuscript which makes it difficult to comment.

We thank the Reviewer for their time and suggestion. We added line numbering in the manuscript.

Below I’m using the page number of the submitted document and the related paragraph.

• P9, par1: ‘one main issues’-> of the

We made the suggested changes in the manuscript (lines: 71 in the manuscript and 78 in the revised

manuscript with trach changes).

• P9, par2: ‘growth, senescence death and litterfall’-> death?

We made the changes in the manuscript (lines: 84 in the manuscript and 91 in the revised

manuscript with trach changes).

• P10, par1: you are mentioning novel approaches, yet the most recent citation is almost a decade

old. Try to incorporate recent advances in the field on this and/or related domains. This is also

something that is missing from the discussion.

We made the suggested changes in the manuscript (references: 56-60).

• P13, par2: remove ‘with’

We made the suggested changes in the manuscript (lines: 169 in the manuscript and 179 in the

revised manuscript with trach changes).

• P13, par2: Please check syntax of the last sentence.

We made the suggested changes in the manuscript (lines: 173-175 in the manuscript and 184-187 in

the revised manuscript with trach changes).

• P18, par4: three main models are stated but four are described (moisture, radiation, temperature,

nitrogen). Also, how are extreme events defined and quantified?

We made the suggested changes in the manuscript (lines: 304 in the manuscript and 328 in the

revised manuscript with trach changes). Extreme events are not quantified itself, but are implicit in

some of the models. As weather data is an input of some models related to vegetation and, each

type of vegetation has limiting vital parameters (minimum and maximum temperature, radiation

and humidity to live), any event outside these vital parameters will influence the outcome of

several models, as the vegetation will not be able to grow or will die. For example, some models

that are influenced directly or indirectly are the vegetation growth model, vegetation death or

livestock ingestion, among others.

• Figure 6. It’s not clear how these values are computed. Are the default values maintained and in

each graph a different limitation is applied? At what values of the limiting factor and what are the

units?

The model values that limit the growth of vegetation are between 0 and 1. These values have no

units as it is an index that is created after several calculations (Supporting Information S1

Appendix: S2-4 Fig and S2-4 A-C Table). It is calculated taking into account whether the

environmental conditions affecting the vegetation (temperature, radiation, humidity and nitrogen)

are sufficient for it to grow. If the vegetation is able to reach its growth needs through the inputs

from the received environmental conditions, it will grow the maximum possible [Ratio of limiting

factor for vegetation growth = 1], equalling the actual growth to the potential growth. If, on the

other hand, environmental conditions are adverse, the "Ratio of limiting factor for vegetation

growth" will be smaller than 1, reaching 0 when there is no growth. Lines: 315-318 in the

manuscript and 338-341 in the revised manuscript with trach changes.

• Figure 7 You point out differences between maximum potential and maximum actual growth. This

could be better visualized if the same color range was applied on both graphs. If the range was 0-

5.29g on both graphs then we’d see a shift of B graph towards yellow/red.

We made the suggested changes in the manuscript (lines: 359-362 in the manuscript and 385-388 in

the revised manuscript with trach changes).

• Figure 8: Caption needs to be more informative and/or translate the Spanish title (and axes). Lines

are ingestion and bars vegetation? What do the error bars stand for?

We made the suggested changes in the manuscript (lines: 370-380 in the manuscript and 403-413 in

the revised manuscript with trach changes).

• Figure 9: Consider the same color range suggestion as figure 7. Also, why is the livestock mass

decreasing? Maybe that’s worth commenting on.

We change the color in the manuscript (lines: 388 in the manuscript and 423 in the revised

manuscript with trach changes). The model has been generated for time 1, the livestock have not

yet been able to eat enough to compensate for the energy they need, so the weight is negative. They

lose more weight in areas with steeper slopes, where less vegetation grows and the effort to move is

greater. Moreover, for this case study, in the majority of plots, cows and mares coexist, creating a

greater competition between them.

• Table 3: please inset units for the parameters.

The parameters in the Table 3 have no units. “IDMancha”, “com”, “com2” and “t” are qualitative

(the numbers are codes related to the variables). The parameters “FT”, “FR”, “FH” and “FN” are

rankings valued 0 to 1. We clarified this in the manuscript (lines: 327-342 in the manuscript and

350-365 in the revised manuscript with trach changes).

• Figure 13: A description of what A and B is should be provided in the caption

Figure 13 have changed to Figure 14. We made the suggested changes in the manuscript (lines:

435-438 in the manuscript and 476-479 in the revised manuscript with trach changes).

• Supporting information

o I’ve found the units for many of the described variables either inconsistent, missing or confusing.

Please carefully go through the whole list and check for inconsistencies and complete missing

information. if the same type of variables have indeed different units please clarify why. Some

examples:

o Proportions: in same cases the range ([0 1]) is given (eg adic, st), some are unitless (eg fhs,

cirhift02) and some have units day (eg pmin, pimx).

We made the changes in the Supporting Information S1 Appendix file.

o Mass: shouldn’t all variables related to contents, biomass, mass be in kg (eg ihcift)? Similar for

concentrations (eg nogm2ift)?

Puerto has different units for mass, volume or area depending on the variable and transform the

units in the code accordingly. For example, in the line of R code from the "Port" model:

, where “VarPesoInd” (livestock weight change) is in grams and PVr (livestock weight) in in kg. In

the case of PaL, this calculation is not necessary because k.LAB does the transformation

automatically.

o Temperature: some missing (eg st1)

We made the suggested changes in the Supporting Information S1 Appendix file.

o Shouldn’t digestibility be in % (eg digm, digv)?

The values are in fact proportions, that is why we have put [0-1] in units instead of the symbol %.

o Table S9. Almost all units are missing.

We made the suggested changes in the Supporting Information S1 Appendix file.

o For pv weight is given in kg but right below for pvr it has no units.

We made the suggested changes in the Supporting Information S1 Appendix file.

o Table 10. What is MJ/t (eg emi), is it different to MJ/d (eg nmd2)?

The Puerto manual uses the symbol "t" in order to identify each time step in the model, which in

this case it refers to day (in Puerto as "d"). At the time of rewriting and adapting part of the Puerto

manual, we have made a typo error as those symbology is confusing and is not using SI

conventions.Energy in some cases it is given, in some omitted (eg enigan, enimd) and in same it is

confusing (eg in egkg, if it’s the energy needed to increase 1kg of body weight then the unit should

be simply MJ).

We made the suggested changes in the Supporting Information S1 Appendix file. “egkg” means

energy needed for increase one kilogram, for this reason the units are MJ/kg.

o Since you are using m^2, and m^3, then volume variables (eg dri, et0) should also be mm^3, not

mm.

Milimeters (mm) are describing a density of a volume over area (mm3/mm2). It is used in incident

volumes representing water content on an area as, for example, precipitation over an area; but it

cannot represent water content in the soil. In the case of water content in the soil, the unit m3/m3 is

refers to a given volume of water contained in given volume of substrate, as the water content at

field capacity or the water content at wilting point1. However, k.LAB can corrects and rescales

units based on spatial and temporal context in case that there is no consistency between models.

(lines: 521-523 in the manuscript and 562-564 in the revised manuscript with trach changes).

1FAO: Allen RG, Pereira LS, Raes D, Smith M. Crop evapotranspiration - Guidelines for

computing crop water requirements - FAO Irrigation and drainage paper 56. FAO (Food and

Agriculture Organization of the United Nation); 1998. Available from:

http://www.fao.org/3/x0490e/x0490e00.htm#Contents; page 162, equation 82 and 83, where ADT

is as TAW and AFA is as RAW.

o Also please check numbering on figure S10

We made the suggested changes in the Supporting Information S1 Appendix file.

---

## [Decision Letter · Decision Letter 2]

15 Dec 2022

PONE-D-22-04207R2Scientific modelling can be accessible, interoperable and user friendly: A case study for pasture and livestock modelling in SpainPLOS ONE

Dear Dr. Marquez Torres,

Thank you for submitting your manuscript to PLOS ONE. After careful consideration, we feel that it has merit but does not fully meet PLOS ONE’s publication criteria as it currently stands. Therefore, we invite you to submit a revised version of the manuscript that addresses the points raised during the review process.

ACADEMIC EDITOR: Please insert comments here and delete this placeholder text when finished. Be sure to:Indicate which changes you require for acceptance versus which changes you recommendAddress any conflicts between the reviews so that it's clear which advice the authors should followProvide specific feedback from your evaluation of the manuscriptPlease ensure that your decision is justified on PLOS ONE’s publication criteria and not, for example, on novelty or perceived impact.

We look forward to receiving your revised manuscript.

Kind regards,

Ji Chen, PhD

Academic Editor

PLOS ONE

Journal Requirements:

Additional Editor Comments (if provided):

Reviewers' comments:

Reviewer's Responses to Questions

**Comments to the Author**

1. If the authors have adequately addressed your comments raised in a previous round of review and you feel that this manuscript is now acceptable for publication, you may indicate that here to bypass the “Comments to the Author” section, enter your conflict of interest statement in the “Confidential to Editor” section, and submit your "Accept" recommendation.

Reviewer #2: All comments have been addressed

Reviewer #4: (No Response)

Reviewer #5: (No Response)

2. Is the manuscript technically sound, and do the data support the conclusions?

Reviewer #2: Yes

Reviewer #4: Yes

Reviewer #5: (No Response)

3. Has the statistical analysis been performed appropriately and rigorously? 

Reviewer #2: Yes

Reviewer #4: Yes

Reviewer #5: (No Response)

4. Have the authors made all data underlying the findings in their manuscript fully available?

Reviewer #2: Yes

Reviewer #4: No

Reviewer #5: (No Response)

5. Is the manuscript presented in an intelligible fashion and written in standard English?

Reviewer #2: Yes

Reviewer #4: Yes

Reviewer #5: (No Response)

6. Review Comments to the Author

Reviewer #2: All my previous concerns and suggestions have been successfully addressed, so the paper is now ready for publication.

Reviewer #4: Torres et al. improve the Puerto model by k.IM language and provide a case study on pasture and livestock modelling in Spain. This work is timely and useful for model users. Because of complexity in model framework and practical application, it would be helpful if the authors can further provide detailed tutorial for end-users, such as user guide or video tutorial. I hope that more modelers will contribute to this model framework based on your nice open-source work. Some specific comments may help improvement of this manuscript.

L264-268, please clarify more information on data sources of PaL models, including local measurements and global data. Is raster as the raw measurements or produced by measurement data by scaling up? And, which global databases did you use?

L280, it is better to clarify the algorithms PaL called. If not, the core of PaL models is unclear and a “black box”.

L290, to what extent the PaL results can directly match the final Puerto outputs? How can we evaluate it? Did you have direct comparison?

L315, how did you calculate the ratio of limiting factors for vegetation growth?

L326 and L407, I cannot find Table 3.

L370, it seems that the legend on the top is not showed in English.

Reviewer #5: (No Response)

7. PLOS authors have the option to publish the peer review history of their article (what does this mean?). If published, this will include your full peer review and any attached files.

Reviewer #2: No

Reviewer #4: No

Reviewer #5: No

---

## [Author Response · Author response to Decision Letter 2]

13 Jan 2023

Review Comments to the Author

Reviewer #4: Torres et al. improve the Puerto model by k.IM language and provide a case study on pasture and livestock modelling in Spain. This work is timely and useful for model users. Because of complexity in model framework and practical application, it would be helpful if the authors can further provide detailed tutorial for end-users, such as user guide or video tutorial. I hope that more modelers will contribute to this model framework based on your nice open-source work. Some specific comments may help improvement of this manuscript.

We thank the reviewer for this suggestion. We added a readme file with the instructions to run the PaL models in k.LAB in the online repository and explained it in the manuscript. (Revised Manuscript with Track Changes: line 183, Manuscript: line 183)

L264-268, please clarify more information on data sources of PaL models, including local measurements and global data. Is raster as the raw measurements or produced by measurement data by scaling up? And, which global databases did you use?

We thank the reviewer for this suggestion. Most of the input data were created within the ALICE project partners (lines 111-117) and it is described in article (1), for instance, the data related to climate and topography. To create the vegetation cover map by species, we validated the biodiversity data and habitat mapping from (1) with data collected by Dr. Juan Busqué from previous projects (2–4). We also used the global soil texture map from the ISRIC (World Soild Information) foundation because we could not develop a local soil map (5). Most of the raster maps were produced through downscaling techniques to improve the spatial resolution. In addition, k.Explorer shows all the data sources used for each model (lines 435-441 and 448-454). We made the suggested changes in the manuscript (Revised Manuscript with Track Changes': lines 266-268, Manuscript: lines 266-268).

1. Fonseca A, Santos JA, Mariza S, Santos M, Martinho J, Aranha J, et al. Tackling climate change impacts on biodiversity towards integrative conservation in Atlantic landscapes. Global Ecology and Conservation. 2022 Oct 1;38:e02216. 

2. Bedia J, Cabañas S, Busqué J. Productivity and plant diversity are related to a community functional signature in mountain grasslands. In 2009. p. 79–82. 

3. Busqué J. De la investigación a la práctica: herramientas para gestionar la ganadería de montaña y los pastos comunales de Cantabria dentro de la política agraria común. Revista Pastos. 2014 Jun;6(1):6–42. 

4. Busqué J, Fernández N, Fernández B. A decision support tool to design rangeland sustainable grazing systems. Sustainable grassland productivity: Proceedings of the 21st General Meeting of the European Grassland Federation, Badajoz, Spain, 3-6 April, 2006. 2006;682–4. 

5. Batjes NH, Ribeiro E, van Oostrum A. Standardised soil profile data to support global mapping and modelling (WoSIS snapshot 2019). Earth System Science Data. 2020 Feb 10;12(1):299–320. 

L280, it is better to clarify the algorithms PaL called. If not, the core of PaL models is unclear and a “black box”.

We made the suggested changes in the S1 Appendix, specifically in the tables named as S”n”C Table Equations of “x”. We add all references and the page number of each below:

• S2C Table Equations of moisture namespace. Page: 5

• S3C Table Equations of radiation namespace. Page: 8

• S4C Table Equations of temperature namespace. Page: 10

• S5C Table Equations of vegetation growth namespace. Page: 13

• S6C Table Equations of vegetation growth namespace. Page: 15

• S7C Table Equations of litterfall namespace. Page: 17

• S8C Table Equations of ingestion namespace. Page: 23

• S9C Table Equations of excretion namespace. Page: 26

• S10C Table Equations of livestock mass namespace. Page: 30

• S11C Table Equations of nitrogen cycle namespace. Page: 34

L290, to what extent the PaL results can directly match the final Puerto outputs? How can we evaluate it? Did you have direct comparison?

We tackled this in the S2 Appendix (Supporting Information, Pages 36-41). We compared the results of Puerto and PaL for the same area, the methodology and the respective limitations.

L315, how did you calculate the ratio of limiting factors for vegetation growth?

We described the calculations in Appendix S1 (Page 8), in the table S3C Table Equations of radiation namespace, in the row with the identifier (id) named ftrh. We clarified in the manuscript (Revised Manuscript with Track Changes: lines 306-309, Manuscript: lines 306-309).

L326 and L407, I cannot find Table 3.

Table 2 was wrongly numbered, thanks for pointing this out (Revised Manuscript with Track Changes: line 330, Manuscript: line 330).

L370, it seems that the legend on the top is not showed in English.

We thank the Reviewer for their time and suggestion. Puerto model displays results as images with legend in Spanish by default. We showed the original Puerto results. We clarified and added the translation in the Figure description (Revised Manuscript with Track Changes: lines 376-382, Manuscript: lines 375-381) as well as in the text (Revised Manuscript with Track Changes: lines 372-373, Manuscript: lines 372-373).

---

## [Editor Report · Decision Letter 3]

23 Jan 2023

Scientific modelling can be accessible, interoperable and user friendly: A case study for pasture and livestock modelling in Spain

PONE-D-22-04207R3

Dear Dr. Marquez Torres,

We’re pleased to inform you that your manuscript has been judged scientifically suitable for publication and will be formally accepted for publication once it meets all outstanding technical requirements.

Kind regards,

Ji Chen, PhD

Academic Editor

PLOS ONE
---

## [Editor Report · Acceptance letter]

26 Jan 2023

PONE-D-22-04207R3 

Scientific modelling can be accessible, interoperable and user friendly: A case study for pasture and livestock modelling in Spain 

Dear Dr. Marquez Torres:

I'm pleased to inform you that your manuscript has been deemed suitable for publication in PLOS ONE. Congratulations! Your manuscript is now with our production department. 

Kind regards, 

on behalf of

Dr. Ji Chen 

Academic Editor

PLOS ONE